# Using R in hydrology: a review of recent developments and future directions

Louise J. Slater[a], Guillaume Thirel[b], Shaun Harrigan[c], Olivier Delaigue[b], Alexander Hurley[d], Abdou Khouakhi[e], Ilaria Prodoscimi[f], Claudia Vitolo[c], and Katie Smith[g]

[a]School of Geography and the Environment, University of Oxford, OX1 3QY, UK
[b]IRSTEA, HYCAR Research Unit, 1 rue Pierre-Gilles de Gennes, 92160 Antony, France
[c]Forecast Department, European Centre for Medium-Range Weather Forecasts (ECMWF), Shinfield Park, Reading, RG2 9AX, UK
[d]School of Geography, Earth and Environmental Sciences, University of Birmingham, B15 2TT, UK
[e]School of Architecture, Civil and Building Engineering, Loughborough University, Loughborough, UK
[f]Department of Environmental Sciences, Informatics and Statistics, Ca' Foscari University of Venice, Venice, Italy
[g]Centre for Ecology & Hydrology, Maclean Building, Crowmarsh Gifford, Wallingford, OX10 8BB, UK

**Correspondence:** Louise J. Slater (louise.slater@ouce.ox.ac.uk)

**Abstract.** The open-source programming language R has gained a central place in the hydrological sciences over the last decade, driven by the availability of diverse hydro-meteorological data archives and the development of open-source computational tools. The growth of R's usage in hydrology is reflected in the number of newly published hydrological packages, the strengthening of online user communities, and the popularity of training courses and events. In this paper, we explore the benefits and advantages of R's usage in hydrology, such as the democratization of data science and numerical literacy, the enhancement of reproducible research and open science, the access to statistical tools, the ease of connecting R to and from other languages, and the support provided by a growing community. This paper provides an overview of a typical hydrological workflow based on reproducible principles and packages for retrieval of hydro-meteorological data, spatial analysis, hydrological modelling, statistics, and the design of static and dynamic visualizations and documents. We discuss some of the challenges that arise when using R in hydrology and useful tools to overcome them, including the use of hydrological libraries, documentation and vignettes (long-form guides that illustrate how to use packages); the role of Integrated Development Environments (IDEs); and the challenges of Big Data and parallel computing in hydrology. Last, this paper provides a roadmap for R's future within hydrology, with R packages as a driver of progress in the hydrological sciences, Application Programming Interfaces (APIs) providing new avenues for data acquisition and provision, enhanced teaching of hydrology in R, and the continued growth of the community via short courses and events.

## 1 Introduction: the rapid rise of R in hydrology

In recent decades, the hydrological sciences, like many other disciplines, have witnessed major changes due to the growth of diverse data archives and the development of computational resources. Hydrology has benefited from the increase in publicly-accessible data, including: (a) observational river flow archives (Hannah et al., 2011) such as the World Meteorological Or-

ganization's Global Runoff Data Centre, which currently includes more than 9,500 stations from 161 countries; (b) gridded reanalysis climate data products such as Copernicus's ERA-Interim (Dee et al., 2011) or ERA5 (Hersbach et al., 2018; Copernicus Climate Change Service, 2018); (c) measurements from sensors and satellites, such as total water storage variations from the Gravity Recovery and Climate Experiment (GRACE, Tapley et al., 2004) or snow cover area from NSIDC MODIS (Hall and Riggs, 2016); and (d) catchment attributes such as the Global Streamflow Indices and Metadata Archive (Do et al., 2018), or national datasets such as GAGESII (Falcone, 2011) and CAMELS (Addor et al., 2017). Together, these datasets have facilitated the investigation of many catchments as well as interdisciplinary research linking meteorology, climatology, hydrology and the earth sciences.

In addition to the availability of large-scale data archives, the increase in computational power and uptake of programming languages have also been a major driver of change in the discipline. Increasingly, hydrologists are using data science approaches to derive process insights from large and complex datasets (e.g. Guo, 2017). The ability to explore and mine these datasets has facilitated a move from in-depth experiments in single catchments towards large-sample studies (e.g. Villarini et al., 2009; Berghuijs et al., 2014; Slater et al., 2015; Archfield et al., 2016; Blöschl et al., 2017; Blum et al., 2017; Harrigan et al., 2018). By applying an analysis to many catchments, or grouping those with similar characteristics, hydrologists have been able to test the broad applicability of hydrological theories and draw systematic insights about hydrological processes (e.g. Blöschl et al., 2013). Large-sample studies are increasingly being employed to develop novel (conceptual and/or physical) models that are applicable across diverse catchment conditions, thereby improving process understanding and leading to more robust predictions (e.g. Gupta et al., 2014). At the same time, a broad range of well-established lumped (e.g. GR4J, Perrin et al., 2003) and semi-distributed (e.g. TOPMODEL, Beven, 1997) models of the hydrological cycle - many of which were initially developed in older languages such as Fortran - have now been incorporated into R packages. Some of them have been translated into R and others remain in their initial languages, e.g. SWAT (Neistsch et al., 2005). The uptake of empirical, conceptual and physically-based computational hydrological models as well as the growth of large-sample studies have driven the emergence of a field that can be described as *computational hydrology* (e.g. Hutton et al., 2016; Melsen et al., 2017): a scientific paradigm that allows hydrologists to explore hydrological questions with a computational approach. Computational hydrology aims to develop reproducible hydrological analyses that can be applied to many catchments, with the aim of generating novel and systematic process insights that improve our physical understanding of the hydrological cycle. The field differs somewhat from both *hydroinformatics* (e.g. Abbott et al., 1991), which has a greater focus on information and communications technologies and/or black box approaches to address water-related issues, and *geocomputation*, (e.g. Openshaw and Abrahart, 2000; Lovelace et al., 2019), which uses similar computational techniques for spatial data analysis.

The growth of computational hydrology has been enhanced by the development of the open-source programming language R, originally developed for statistical computing by Ross Ihaka and Robert Gentleman in the 1990s (Ihaka and Gentleman, 1996; R Core Team, 2018) and supported by an enthusiastic and rapidly-growing online community. As a free multi-platform language, R is highly versatile and has a wide range of uses including: data acquisition and provisioning, manipulation, analysis, modelling, statistics, visualization, and even well-developed geospatial and geographic information system (GIS) applications. R can be used for generating reports, interactive presentations for teaching or conferences, or even to prototype dashboards

and web applications. One of the greatest strengths of R is its extremely active community of users, who, in the past 25 years, have developed and released in the public domain more than 10,000 packages spanning many scientific disciplines. The Comprehensive R Archive Network (CRAN, https://cran.r-project.org) is the main repository for these packages, hosted by a network of ftp and web servers around the world. The abrupt increase in the number of hydrological package updates in
5   2018 exemplifies the growing importance of the language in hydrology and strengthening of the community (Figure 1). These developments have allowed the R language to fit comfortably in production-ready ecosystems (e.g. web applications) and take advantage of cutting-edge technologies and tools for improving reproducibility, testing and continuous integration.

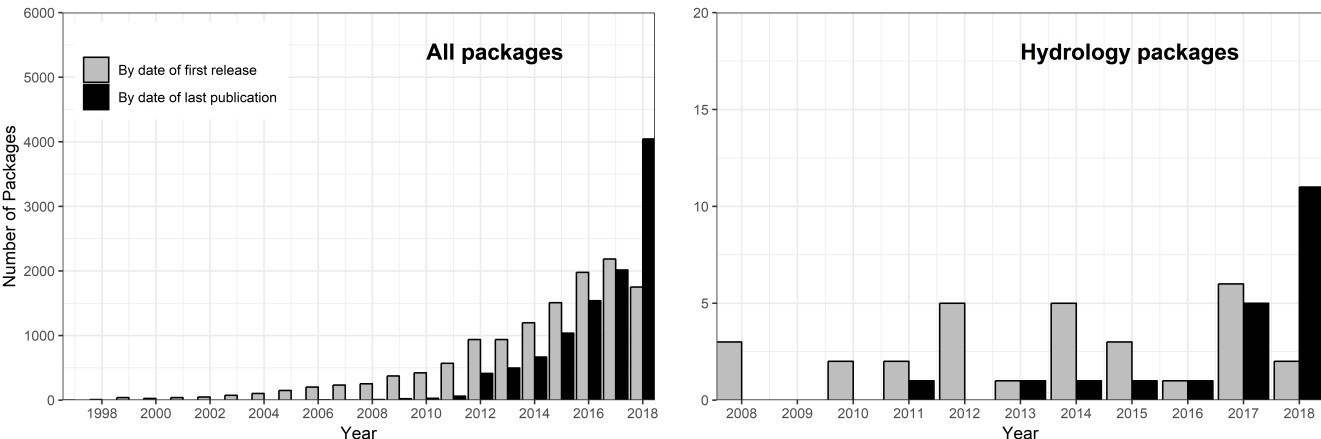

**Figure 1.** The number of R packages available on CRAN (1997-2018). Left: all packages; right: hydrology packages only (defined as those that include the string "hydro" within the package metadata; erroneous packages containing terms like hydrocarbon were removed). Bar colors indicate number of packages published based on (1) date of first release (grey bars; https://cran.r-project.org/src/contrib/archive); (2) date of last publication - i.e. most recent publication date of packages that are currently available on CRAN (black bars; https://cran.r-project.org/web/packages/available_packages_by_date.html). In both cases each package is counted only once; for example the package `hydroTSM` was first released in 2010 (grey) but its most recent update is 2017 (black). The script to produce this figure is provided as part of the supplementary materials (Supplement 1). Data were downloaded from CRAN on 4 June 2019.

This paper aims to provide a broad overview of the utility of R in the hydrological sciences and of important developments in recent years. In Section 2 we provide a summary of the many benefits and advantages of using R in hydrology. In Section 3
10   we describe some of the key hydrological packages that have been developed by the community, as well as general packages of broad application in the sciences. In Section 4 we discuss some of the challenges that the R-Hydro community faces, including tools and solutions to overcome these challenges. Finally, in section 5 we list some of the future directions to strengthen the computational hydrology community.

## 2 The benefits and advantages of using R in hydrology

### 2.1 Democratizing open science and numerical literacy

One of the principal advantages of R is its ease of use, resulting from typically detailed documentation, a large number of online resources, object-oriented programming (the language is organized around objects with unique attributes), functional

programming (the code can be written with functions to facilitate modularity and avoid changing-state data), and the availability of the source code under the open-source license. Further, R can be run on all major operating systems (i.e., Microsoft Windows, macOS, Linux), making it ideal for institutional or personal use. In contrast with compiled languages, such as C or Fortran, R is an interpreted language, which means that the code can be written and executed line-by-line. In practice this means that achieving a basic hydrological analysis can be as simple as writing a sequence of commands to read a file, clean the data, and

plot a graph. These sequences of commands are typically collated in an "R script"; together, multiple scripts may constitute a hydrological workflow (see Figure 3). A high level of documentation and support exists for each of the tasks within a hydrological workflow, including a diverse range of packages (see Section 3). R-Users may search for R-relevant content (such as expressions, packages and functions) using the dedicated internet search engine, RSeek (https://rseek.org), as well as generalist search engines. For many R newcomers, the initial hurdle is finding the time required to gain basic understanding of

R itself; but these efforts pay off rapidly as one learns how to find solutions on the Internet (see section 2.5) and to navigate help files provided within R.

One of the advantages of writing a script is that it can be reused and improved incrementally over time, so the author never has to repeat a task manually (in contrast to point-and-click software that lack a programmatic interface). Moreover, the same script can be re-used repeatedly by different people to reproduce a given result (avoiding duplication of efforts) or to analyze

different data: e.g. different catchments, different types of gridded data, or the same data at different points in time. Thus, scripts written in R (or other languages) have wide-ranging benefits, as they facilitate the testing and quality-control of the scientific workflow, can be shared and improved by a team of users, lessen the risk of making manual mistakes, and significantly enhance the speed with which analyses can be conducted and updated.

Recent developments in R have contributed to enhancing open science and numerical literacy in the hydrological sciences.

R's ease of access and use has improved what might be termed "scientific computing literacy" within the hydrological community. Volunteer projects such as the Software Carpentry (Wilson, 2006, 2014) have been teaching basic computing skills to researchers since 1998, and R now forms a central part of their training. Over the past decade or so, R has become one of the core tools for scientific computation in hydrology. Hosted instances of R allow the user to run R and RStudio® (an IDE, described in greater detail in section 4.2) in the cloud, i.e. in a web browser rather than locally on one's own computer. These

hosted instances have made the language more accessible to non-specialists, due to the large range of pre-installed packages. Importantly, the RStudio Cloud (https://rstudio.cloud/) has recently been developed "to make it easy for professionals, hobbyists, trainers, teachers and students to do, share, teach and learn data science using R", and provides many learning materials, including interactive tutorials covering the basics of data science, cheatsheets for working with popular R packages, links to

DataCamp® courses, and a guide to using RStudio Cloud. Such hosted instances of R remove the initial hurdle of installing R and the required packages (i.e. the technicalities), and ultimately make scientific work more accessible.

The hydrologic R-Users (R-Hydro) community has developed a number of platforms to share computational hydrology analyses, and code is increasingly being shared via repositories (see section 5.1). Code and results can be published as traditional media (e.g. as articles, supplemental material, scripts, packages, or computational environments) or on the web via blog posts, snippets or tutorial documents, allowing users to engage interactively. The "literate programming" paradigm (i.e., interspersing snippets of code and written text within a unique document; see Knuth, 1984) has also become increasingly popular due to the development of dashboards and online publishing platforms. Dashboards are interfaces that may include a group of related data visualizations such as charts, graphs, tables and maps (see e.g. Figure 6). They can be fixed (e.g. a non-interactive web page) or dynamic (e.g. allowing users to alter input values and see how outputs change virtually in real-time). Hydrologists have increasingly employed dashboards to display their analyses in an interactive, user-friendly manner, using packages such as `plotly`, a graphing library that makes publication-quality graphs online (Sievert, 2018), the `shiny` web application framework (Chang et al., 2015), `shinydashboard` (Chang and Borges Ribeiro, 2018), or the `flexdashboard` package (Iannone et al., 2018). Additionally, free services such as RPubs (a free web-publishing platform, https://rpubs.com) and the aforementioned `plotly` enable the publishing of these static or interactive documents and visualizations online. In doing so, these publishing services facilitate immediate interaction and knowledge transfer.

## 2.2 Enhancing reproducible hydrological research and open science

Reproducibility is a key feature of the scientific method and can be broadly defined as the ability for the community to reproduce and verify previous findings. Encouraging reproducible practices helps reduce the likelihood of errors (methods can be tested by other researchers), while increasing the uptake of any positive developments within a discipline (Hutton et al., 2016). However, scientific research is increasingly under fire for its lack of reproducibility due to missing or inadequate methodology description (Ceola et al., 2015) and model and data availability. True reproducibility requires more than the mere repeatability of results with the same computer code and data: one must also be able to reproduce a study's conclusions when testing the theory with different data or a different model setup (Melsen et al., 2017).

The open-source nature of R packages and the CRAN repository set-up are one key added value of R to reproducible geoscientific research (Pebesma et al., 2012). The CRAN repository ensures the traceability of past analyses by archiving former versions of the packages compiled on any platform (https://cran.r-project.org). In addition, packages are citeable in reports and papers together with their version number, allowing the user to track which code was used. The package developers are also "traceable" on the CRAN via their ORCID number, which provides an indication of whether they also authored any corresponding scientific papers. The CRAN Task Views provide guides to the packages and tools that exist on CRAN for the different disciplines. Note that many packages are submitted to on-line repositories such as GitHub that never make it to CRAN for a range of reasons, but this does not necessarily make them less reliable. The Task Views provide a list of tools which can enhance reproducibility in R (see https://cran.r-project.org/web/views/ReproducibleResearch.html). Relying on well-established publishing platforms such as CRAN and GitHub has promoted the use of best practice in writing R code.

Packages such as `rhub` (Csárdi, 2017), while packages like `roxygen2` help users create documentation for their R packages (Wickham et al., 2018a). Other initiatives, such as rOpenSci (carefully vetted scientific R software tools, see https://ropensci.org), and their code peer-review, have facilitated the implementation of best practices whilst holding authors to scientific scrutiny.

Journals in the field of hydrology such as *Hydrology and Earth System Sciences*, *Water Resources Research*, and the ASCE journals *Journal of Water Resources Planning and Management* and *Journal of Hydrologic Engineering* now actively encourage authors to publish the data and computer codes underlying the results presented in their papers. The journal *Nature* states that a manuscript can be rejected if the code used to generate new analyses cannot be provided to the editors/reviewers. Despite the advantages of sharing hydrological code, few computational hydrologists do this because cleaning and annotating the code places an additional burden on the publishing timeframe. However, it is reasonable to assume that, as more journals require submission of codes with papers, the community of computational hydrologists (and associated fields) will continue to grow and strengthen the field. In addition to the sharing of open-source code, reproducibility experts also advocate the use of software toolsets such as version control, scripting, container technology, and computational notebooks to enhance the reproducibility of scientific results (Perkel, 2018). Hydrological tutorials, vignettes or teaching documents increasingly implement literate programming (Section 2.1), where the code and results are described in plain English within one same document or webpage.

In the hydrological sciences, several ongoing open scientific initiatives can be noted, other than the R packages that are discussed in Section 3. These initiatives include the HydroShare web-based system for sharing hydrologic data and models, which allows hydrologists to visualize, analyze, and work with data and models on the HydroShare website (e.g. creating and publishing a Web App resource; Essawy et al., 2018). Additionally, the Sharing Water-related Information to Tackle Changes in the Hydrosphere for Operational Needs (SWITCH-ON) virtual laboratory aims to explore the potential of Open Data for water security and management (Ceola et al., 2015). The reproducibility agenda has benefited from strong political and financial support, with European Union projects like FOSTER Plus (Fostering the practical implementation of Open Science in Horizon 2020 and beyond) that aim to encourage open science. The European Commission's "Science with and for Society" program, implemented as part of Horizon 2020 (European Commission, 2015, p. 17) similarly noted that the re-use of research data generated with public funding should have beneficial impacts for science, the economy and society. As such, open science is expected to encourage interdisciplinary research, and is considered a key approach to tackle the grand challenges of our time.

## 2.3 Providing statistical tools for hydrology

There are many different types of software for statistical analysis, but R is still considered the most powerful and popular language and environment for statistical computing. For example, a search for the word "statistics" on Stack Overflow, the question and answer website for programmers, on 18 May 2019 generated 12,339 results with the tag "R", 9,582 results with Python, 4,916 with Java, 2,623 with C#, and 48 with Fortran. R is "GNU S", a language which can be described as the modern implementation of the S language (Becker and Chambers, 1984; Chambers, 1998) and is specifically optimized for statistical computing. In addition to the standard statistical techniques within base-R (i.e., the in-built basic functions that define R as a

language), R also provides access to a large variety of advanced and recent statistical packages, which have been developed by its user community of statisticians and statistically-minded scientists working in a range of research fields. When comparing R with other similar open-source languages, users often describe R's unique selling point as the vast number of statistical packages which liken it to a free, community-driven version of statistical software.

The statistical and graphical packages provided in R are particularly useful for the hydrological sciences, and include techniques such as linear and nonlinear modelling, statistical tests, time series analysis, classification, or clustering. A description of specific statistical packages relevant to hydrology is provided in section 3.6.

## 2.4    Connecting R to and from other languages

Many different programming languages are used in hydrology, including older languages such as Fortran, developed in the
1950s (e.g. Backus, 1978) and C in 1972 (e.g. Ritchie, 1993); proprietary commercial languages e.g. MATLAB®, initially released in 1984 (MathWorks, 2018); and more recent open-source languages like Python, which appeared in 1990 (e.g. Sanner et al., 1999; Guttag, 2013), R in 1993 (e.g. R Core Team, 2018; Ihaka, 1998), and Julia in 2012 (Bezanson et al., 2012).

S and R were both built using algorithms implemented mostly in Fortran and sometimes in C (Chambers, 2016, p.55), which is why R can be connected natively to both languages. Interconnections between R and other languages can be very useful
for taking advantage of each language's strengths, or using pre-existing scripts that were originally written in other languages. For example, C and Fortran are both efficient in performing loop tasks. This is the reason why the `apply` functions loops are coded in C. The `airGR` hydrological models (Coron et al., 2017, 2018) also make use of Fortran in order to take advantage of its CPU efficiency.

R can also be connected to different languages using a range of packages, e.g. C++ (package `Rcpp`, Eddelbuettel and
Balamuta, 2017) and Java (package `rJava`, Urbanek, 2018). Connections to Python can be achieved with the packages `reticulate` (Allaire et al., 2018b), `rPython` (Bellosta, 2015), `rJython` (Grothendieck and Bellosta, 2012), and `XRPython` (Chambers, 2017). Additionally, R can also be connected to Javascript (e.g. package `V8`, Ooms, 2017), Matlab (e.g. package `R.matlab`, Bengtsson, 2018), or Julia (packages `JuliaCall` and `XRJulia`, Chambers, 2018; Li, 2018). Conversely, connecting other languages to R is also possible. For example, `rpy2` (Gautier, 2018) is the Python interface to the R language and
runs an embedded R, providing access from Python using R's own C-API. The Python library `pyRserve` (Heinkel, 2017) also connects Python with R using Rserve. The `RCall` package allows users to call R from Julia (Bates et al., 2015).

## 2.5    Interacting with the R-Hydro community: scientific resources and courses

One of the major advantages of R is the extensive user community, which provides ample support to newcomers through various initiatives, and is growing at a fast pace. R-Hydro beginners are strongly encouraged to join the discussion on various R-related
topics on social media. On Twitter, the "#rstats" community is particularly active, and users are exposed to communications about a range of new packages and recent developments. Stack Overflow (https://stackoverflow.com) is the go-to online discussion forum for any user, from beginners to expert developers, and provides code snippets to solve a wide range of common problems. In addition, the R project also offers thematic mailing lists (https://www.r-project.org/mail.html), relating to usage

(R-help), package development (R-package-devel) and language development (R-devel), in order to help others, report bugs, or propose solutions. More recent but already extremely active is the RStudio Community forum (https://community.rstudio.com), which includes users interested in RStudio-developed applications and packages.

A wide range of scientific resources, including online manuals and tutorials in several languages, have been developed by the community. The rOpenSci project (https://ropensci.org) brings together a community of volunteers who promote the open development of packages in a non-profit initiative, by reviewing scientific R packages before they are uploaded to CRAN. rOpenSci has contributed notably to the development of best practice through free code reviews and targeted/invited blog posts. Resources can be found in many places online, such as the *Journal of Open Source Software (JOSS)*, which hosts a wealth of short papers documenting R packages.

Short courses, i.e., voluntary training sessions for R users run during conferences like the European Geosciences Union (EGU), have also grown in popularity in the hydrological community in recent years (Figure 2). Since the 2017 EGU General Assembly, the "Using R in Hydrology" short course has been organized in conjunction with the Young Hydrologic Society (YHS, https://younghs.com) as part of the annual Hydrological Sciences division program. The core aim of the short course is to bring together the R-Hydro community, including both users who are learning to program, and advanced users who are either teaching computational hydrology or developing packages for hydrological analysis. Recent advances in R for hydrology applications are demonstrated during the short course and a platform for open discussion between guest speakers and course participants is provided. By rotating topics each year, a valuable repository of slides, code examples, and follow up material is provided to the community on GitHub (https://github.com/hydrosoc; see section 5.6 for further information). It is also worth noting that a regular short course is also run at the EGU on R package development. In addition to the EGU, other regular conferences and meetings such as those run by the International Association of Hydrological Sciences (IAHS) and the Prediction in Ungauged Basins (PUB Summer school) also organise R courses during their meetings. The IAHS Statistical Hydrology (STAHY) commission regularly organizes workshops and short courses on various statistical topics which typically come with a computer-based component in R.

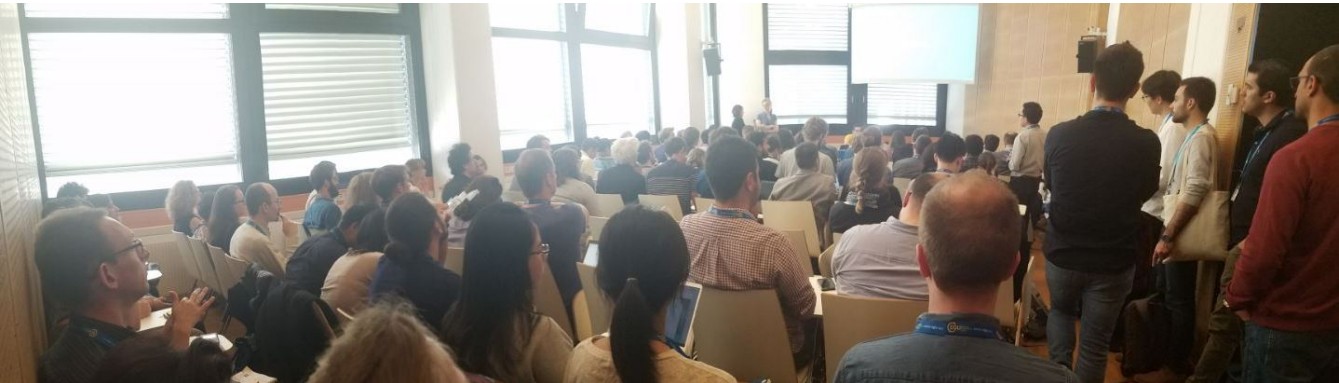

**Figure 2.** EGU Short course "Using R in Hydrology" (https://meetingorganizer.copernicus.org/EGU2018/session/28914). Credit: A. Khouakhi, 11 April 2018.

Finally, it is worth mentioning that the continued growth of the R community is supported by the R Consortium (https://www.r-consortium.org/about), a group with an open-source governance and foundation model that supports the worldwide community of users, maintainers and developers of R software. The R Consortium provides support to the community in multiple ways. One example of such support is the grant program run by the R Consortium's Infrastructure Steering Committee (ISC), which funds development of projects seeking to promote improvement of the R infrastructure and to achieve long term stability of the R community.

## 3 R packages in a typical hydrological workflow

R is an ever-growing environment, as can be seen in the number of R packages that are developed every year (Figure 1). There are now hydrological packages for every step of a standard hydrological workflow (Figure 3); we describe each of these steps in subsequent sections.

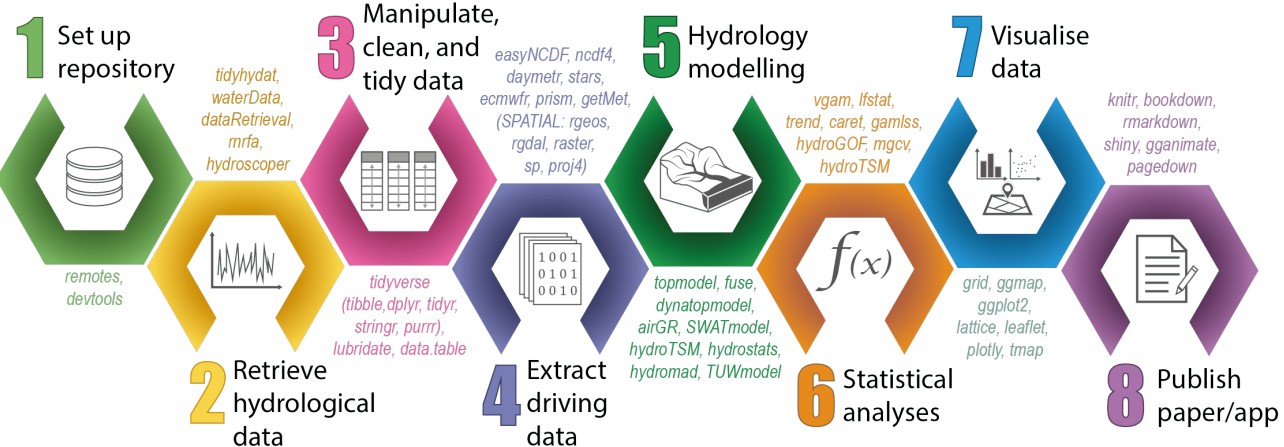

**Figure 3.** A typical hydrological workflow in R, containing eight steps. A selection of relevant R packages used within each script is indicated in coloured text; these packages are described in further detail in sections 3.1 to 3.8.

## 3.1 Setting up a repository and finding the right packages

Setting up a repository with version control (i.e. a reviewable and restorable history) at the start of a research project has many advantages. A repository is a structured set of files that will track edits any team member makes to the project, similar to the "track changes" function in common word processors. In R, version control can be implemented quite simply by connecting RStudio with Git or Subversion, through hosting services like Bitbucket, GitHub, GitLab (which are like a DropBox for code). There are just a few initial steps to set up a repository. These include: (1) creating a local directory (folder) to host your (RStudio) project; (2) creating a Git or a Subversion repository online; and (3) linking the two. Beyond those three steps, all

the user needs to do is regularly commit (i.e. save) their code, together with a very brief summary of the changes made. These changes can then be uploaded, updating and synchronizing the online repository with labelled annotations (user name and timestamp). The repository can be public or private, with different levels of administrator access for users. Many tutorials on how to set up a repository in R can be found online. One example is the Software Carpentry course on "R for Reproducible Scientific Analysis" (https://swcarpentry.github.io/r-novice-gapminder/) which also provides valuable suggestions on how to structure a project repository (e.g. with source data as "read-only" and generated data as "disposable", and folders such as "scr" for scripts and "doc" for documentation).

Once a project folder/repository has been set up, one might need to identify the most useful R packages and functions for the task at hand. CRAN Task Views were recently developed to provide thematic lists of the packages that are most relevant to specific disciplines. The hydrology Task View for "Hydrological Data and Modelling" (https://cran.r-project.org/web/views/Hydrology.html) was published to the CRAN in January 2019 and lists over a hundred packages under the following categories:

- Data retrieval: hydrological data sources (surface or groundwater, both quantity and quality); meteorological data (e.g. precipitation, radiation, temperature, both measurements and reanalysis).

- Data analysis: data tidying (e.g. gap-filling, data organization, quality control); hydrograph analysis (functions for working with streamflow data, including flow statistics, trends, biological indices); meteorology (functions for working with meteorological and climate data); spatial data processing.

- Modelling: process-based modelling (scripts for preparing inputs/outputs and running process-based models); statistical modelling (hydrology-related statistical models).

Additionally, many of the other 38 Task Views that were available in January 2019 were relevant for hydrology (https://cran.r-project.org/web/views). For example, the "EnvironMetrics" Task View contains a Hydrology and Oceanography section, and the "High-Performance and Parallel Computing with R" Task View can also be useful for hydrologists using large amounts of data. Other relevant Task Views cover a range of topics such as time series analysis, machine learning, or spatial analysis.

One last way of discovering relevant and useful hydrological packages is social media, such as Twitter, where many hydrologists share their most recent publications as well as links to useful resources and packages. Some of the Twitter handles that highlight relevant packages for computational hydrology include the USGS group supporting R scientific programming (https://twitter.com/USGS_R) or the rOpenSci page (https://twitter.com/ropensci).

### 3.2 Packages for retrieving hydro-meteorological data

One of the most useful computational advances in recent years has been the development of packages designed specifically to retrieve data from online hydrological archives. Different packages have been designed for importing hydrometric data from repositories such as `dataRetrieval` for the U.S. Geological Survey (USGS)'s National Water Information System (Hirsch and De Cicco, 2015), or `rnrfa` for the UK's National River Flow Archive (Vitolo et al., 2016a). Many of these packages provide vignettes that illustrate how to use the functions. For example, the USGS's `waterData` package allows the user to

import USGS daily hydrological data from the USGS web services, plot time series data, fill in missing values, or visualize anomalies over a range of timescales (e.g. one year, 30 days or one day), in just a few lines of code (see the waterData package vignette on the CRAN; Ryberg and Vecchia, 2017). Other relevant packages are listed in Table 1.

Additionally, many data retrieval packages that are relevant for hydrological analyses have been developed by related scientific disciplines, such as meteorology and climatology. In the future it seems likely that most water and meteorological agencies around the world will facilitate access of these data via APIs and open-source packages (see Section 5.4 for further information on APIs and the possible future of hydro-meteorological data provision). Table 1 is far from exhaustive and there are many other relevant packages that are available on CRAN. For example, the new `ecmwfr` package provides a straightforward interface to the public ECMWF API Web Services and the Copernicus Climate Data Store (CDS), where open datasets such as the ERA5 reanalysis are openly available (Hufkens, 2018).

**Table 1.** Examples of packages for hydrological and/or meteorological data retrieval. See the Hydrology Task View for latest additions https://cran.r-project.org/web/views/Hydrology.html

| Package | Description |
| --- | --- |
| *Hydrological data* | |
| dataRetrieval | Retrieve USGS and EPA hydrologic and water quality data (Hirsch and De Cicco, 2015) |
| hddtools | Hydrological data discovery tools (Vitolo, 2017) |
| hydroscoper | Interface to the Greek national data bank for hydrometeorological information (Vantas, 2018) |
| rnrfa | Retrieve, filter and visualize data from the UK National River Flow Archive (Vitolo et al., 2016a) |
| tidyhydat | Extract and tidy Canadian hydrometric data (Albers, 2017) |
| waterData | Retrieve, analyze, and calculate anomalies of daily hydrologic time series data (Ryberg and Vecchia, 2017) |
| *Climatological data* | |
| daymetr | Interface to the Daymet web services: NASA daily surface weather and climatological summaries over North America, Hawaii, Puerto Rico (Hufkens et al., 2018) |
| ecmwfr | Interface to the public ECMWF API web services (Hufkens, 2018) |
| getMet | Get meteorological data for hydrologic models (Sommerlot et al., 2016) |
| prism | Access the Oregon State Prism climate data using the web service API data (Hart and Bell, 2015) |
| rnoaa | Interface to NOAA weather data (Chamberlain, 2019) |

### 3.3 Packages for reading, manipulating, and cleaning the data

Once data have been retrieved or downloaded, a broad range of packages are available for reading different types of data and their associated metadata. However, these packages are not specifically hydrological and so will be discussed here with brevity.

Note that in many cases, for example with the `dataRetrieval` package, the data are imported directly into the R workspace. Basic data formats such as *csv* and *txt* can be read with base-R (read.table function), or with additional packages that form part of the `tidyverse` package suite (Wickham, 2017), which facilitate reading a range of formats including *xls(x)*, *tsv*, *fwf* (e.g. package `readr`; Wickham et al., 2018b). The `rio` package facilitates data import and export by making assumptions about 5 data structure from the file extension. (Chan et al., 2018).

For reading or writing netCDF files, a number of packages like `ncdf4` (Pierce, 2017), `easyNCDF` (BSC-CNS and Manubens, 2017), `stars` (Pebesma, 2018b), or `raster` (Hijmans, 2017) can be used. GRIB files can also be handled with packages such as `raster` or `gribr` (Wendt, 2018). There is still relatively minimal support for Open Geospatial Consortium (OGC) services in R, although there are some packages such as `sos4R` (of which a new version is expected to be released on CRAN 10 soon), `ows4R` (Blondel, 2018b), or `geometa` (Blondel, 2018a).

Observed hydro-climatological time series data typically need to be "cleaned" because they suffer from various data gaps and errors. For an overview of the different issues with hydrological data see Wilby et al. (2017). This step may involve handling missing data, checking data completeness, reshaping and aggregating data, or converting strings to date format. We do not develop this section specifically because these are general tasks in R. However, for overviews and tutorials on manipulating 15 and cleaning hydrological data we point the reader to published resources from the "Using R in hydrology" workshop (see section 5.6).

## 3.4 Packages for extracting driving data, spatial analysis and cartography

In the past, R may have been a less powerful alternative to the more established spatial software for processing large datasets and extracting information. Now, however, R can be parallelised more easily than other software (harnessing the power of 20 multiple processor cores to handle large datasets) and can integrate GIS analyses within a complete, automated hydrological workflow, which includes data processing steps (before or after any GIS analyses) and any subsequent statistical analyses. It is this integration of GIS as one step within the hydrological workflow that makes R extremely attractive. As a result, in recent years, R has become the "go to" for geocomputation and geostatistics and can now be used as a GIS in its own right. Multiple books have been published on the topic of spatial analysis and mapping with R (Brunsdon and Comber, 2015), or more broadly, 25 geocomputation with R (Lovelace et al., 2019), which includes topics such as reading and writing geographic data and making maps in R.

Many methods are now implemented within R for handling vectorial data, with packages such as `sp` (Pebesma and Bivand, 2005; Bivand et al., 2013) for plotting data as maps or for spatial selection; package `rgdal` (Pebesma and Bivand, 2005; Bivand et al., 2013), which provides bindings to the Geospatial Data Abstraction Library (GDAL) for reading/writing data and 30 access to projection and transformation operations from the PROJ.4 (Urbanek, 2012) library; package `rgeos` (Bivand and Rundel, 2018), which provides an interface to the Geometry Engine Open Source (GEOS) library for geometrical operations; or package `sf` (Pebesma, 2018a), which provides support for simple features to encode spatial vector data. Version 3.0 of ggplot2 offers support to visualize sf objects directly with a specific geom (geom_sf), which allows users to visualize spatial data (raster, shapefiles) easily. There is also a range of packages for handling raster data, like `raster` (Hijmans, 2017), which

**Table 2.** Examples of packages for hydrological modelling.See the Hydrology Task View for latest additions: https://cran.r-project.org/web/views/Hydrology.html

| Package | Description |
|---|---|
| airGR | Suite of GR Hydrological Models for Precipitation-Runoff Modelling (Coron et al., 2017, 2018) |
| brook90 | Implementation of the BROOK90 hydrologic model (Federer, 1995; Schmidt-Walter, 2018; Kronenberg and Oehlschlägel, 2019) |
| dynatopmodel | Implementation of the Dynamic TOPMODEL Hydrological Model (Metcalfe et al., 2015, 2018) |
| Ecohydmod | Ecohydrological Modelling (Souza, 2017) |
| fuse | Ensemble Hydrological Modelling (Vitolo et al., 2016b) |
| hydromad | Hydrological Model Assessment and Development (Andrews et al., 2011; Andrews and Guillaume, 2018) |
| RHMS | Hydrologic Modelling System for R Users (Arabzadeh and Araghinejad, 2018) |
| topmodel | Implementation of the Hydrological Model TOPMODEL in R (Buytaert, 2018) |
| TUWmodel | Lumped Hydrological Model for Education Purposes (Viglione and Parajka, 2014) |

can be used to read, write, manipulate, analyze and model gridded spatial data; and stars (Pebesma, 2018b) for reading, manipulating, writing and plotting spatiotemporal arrays. With all these packages, R now has full cartography and mapping functionality, and can produce sophisticated maps that are either static or interactive. Packages such as cartography (Giraud and Lambert, 2016) or tmap (Tennekes, 2018) can both be used for creating symbols, choropleth or other types of maps, and

5 for facilitating the visualisation of spatial data distributions in thematic maps. These two packages are quite similar in content but not in form: cartography uses the painter's model and while tmap uses the Grammar of Graphics and allows users to draw static and dynamic maps with the same code. Additionally, ggmap (Kahle and Wickham, 2013) is commonly used for visualizing, geolocating and routing spatial data on top of static maps such as Google Maps; RgoogleMaps is used to interact with Google Maps; while the JavaScript leaflet (Cheng et al., 2018) library can be used for creating interactive

web maps. R software can also be linked to a range of other GIS software to take advantage of specific capabilities. R can access ArcGIS geoprocessing tools (ESRI, 2018) by building an interface between R and the ArcPy Python side-package using the RPyGeo package (Brenning et al., 2018b). R can also call wrappers for GDAL/OGR utilities (GDAL Development Team, 2018) using gdalUtils (Greenberg and Mattiuzzi, 2018). There are also interpreted interfaces between the GRASS GIS (GRASS Development Team, 2019) and R from within the GRASS environment or from R, using the packages spgrass6

(Bivand, 2016) or rgrass7 (Bivand, 2018). R can be integrated with QGIS (QGIS Development Team, 2018) using RQGIS (Muenchow et al., 2017), or with SAGA GIS (SAGA Development Team, 2008) using RSAGA (Brenning et al., 2018a).

### 3.5 Packages for hydrological modelling

The next step in a typical hydrological workflow is to conduct hydrological modelling by using the data inputs prepared in previous steps. Hydrological modelling often proceeds by simplifying hydrological processes to test hypotheses about the water cycle, manage water resources, reconstruct incomplete flow time series, predict extreme events (floods or low flows), or to anticipate the effects of future climatic or anthropogenic changes. In Table 2 we highlight some of the key packages that facilitate the implementation of certain hydrological models in R. As R can be used for every step within the hydrological modelling process, from importing and cleaning data, to exploratory analyses, data modelling, data analysis, and graphical visualization, it represents an ideal language for hydrological modellers.

Several well-known hydrological models are provided in these packages, such as the HBV model in the `TUWmodel` package (Viglione and Parajka, 2014); TOPMODEL in the packages `topmodel` (Buytaert, 2018) and `dynatopmodel` (Metcalfe et al., 2015, 2018); SWAT in `SWATmodel` (Fuka et al., 2014); and GR4J in `airGR` (Coron et al., 2017, 2018) and `hydromad` (Andrews et al., 2011; Andrews and Guillaume, 2018). Some packages also include degree-day snow models (e.g. `airGR`, `hydromad` and `TUWmodel`) that are used in nival basins to simulate the accumulation and melt of the snowpack, which greatly impacts the flow regimes. Many of the models included in these packages are lumped (i.e., consider the catchment as a single unit with area-averaged variables) or conceptual (i.e., provide a simplified representation of the physical processes with empirical equations describing the interactions between the processes) models. Some packages, such as `TUWmodel` and `topmodel`, allow a spatial distribution of the hydrological processes. `TUWmodel`, for instance, allows the user to implement a semi-distributed application with differentiation into elevation zones, while `topmodel` allows the user to make topographic index classes from a topographic index map. The FUSE model (Clark et al., 2008) originally implemented in Fortran, is still actively developed and also has a dedicated R package called `fuse` (Vitolo et al., 2016b) that is a re-implementation of the FUSE 2011 model. The vast majority of the models in the above packages are deterministic; however, the `fuse` package can also provide ensembles of conceptual lumped hydrological models and the `hydromad` package is intended to provided a flexible framework to assemble different soil moisture accounting and routing schemes.

The above packages typically allow the user to run the hydrological models, and usually provide some sample input data, with executable examples. Some packages provide optimization algorithms, criteria calculation and dedicated plotting functions (e.g. `airGR` and `RHMS`). Developing hydrological modelling packages in R is relatively straightforward because of the language's flexibility; additionally, as discussed in Section 2.4, R can be connected to other programming languages. This property allows R-users to embed already existing model codes into R and provides a more friendly environment, as many packages ease the analysis of data and model simulations.

### 3.6 Packages for hydrological statistics

R was initially developed as a statistical computing language and is still the primary language in which novel statistical methods are coded and distributed. Statistical approaches are employed for an extremely wide range of tasks in hydrology and it is virtually impossible to give a complete coverage of all possible packages that might be useful to hydrologists. The `skimr`

package provides compact and flexible summaries of data; can be used with pipes and displays nicely in the console. Many estimating procedures can be carried out using the base-R `stats` package, which includes for example correlation analysis and Mann-Kendall testing, linear regression, Poisson regression and Gamma regression. There are also generalist packages for non-parametric trend tests such as `trend` (Pohlert, 2018), which includes a broad range of tests for trend detection, correlation,

as well as detection of change-points or non-randomness. Many general modelling packages have been found to be useful by hydrologists, for example `mgcv` (Wood, 2017), `VGAM` (Yee et al., 2015) and `GAMLSS` (Rigby and Stasinopoulos, 2005), which implement generalized additive regression models for a large number of distributional families. While `mgcv` can be useful for more complex smoothing approaches (e.g. tensor product smooths or varying coefficient models), the focus of `GAMLSS` is to develop flexible models for a large class of distributions which are not part of the exponential family of distributions,

allowing for parameters other than the location to be modelled as a smooth function of the covariates of interest. A data analyst might therefore choose which package to use depending on the most likely distribution of the data which are being analyzed and the complexity of the analysis carried out. The `caret` package (Kuhn et al., 2018) is more commonly used for predictive modelling and machine learning, as it allows the user to develop an impressive range of predictive models (e.g. neural networks, deep learning and decision trees).

There are many packages available for common hydrological tasks. A comprehensive set of functions for carrying out Extreme Value Analysis (Coles, 2001) can be found in the `extRemes` package (Gilleland and Katz, 2016), and a more complete overview can be found in the CRAN "Extreme Value Analysis" Task View. `nsRFA` consists of a collection of statistical tools for objective (non-supervised) applications of the Regional Frequency Analysis methods in hydrology (Viglione, 2018). Lmoments can be implemented using the `lmom` Hosking (2019b) package, while the `lmomRFA` package Hosking

(2019a) enables regional frequency analysis. The `lfstat` (Koffler et al., 2016) package provides functions to compute statistics and plots for low flows similar to those in the manual on low-flow estimation and prediction (WMO, 2009). Many of the standard model evaluation metrics are provided directly within these packages. The `caret` package, for instance, imports other packages, such as `ModelMetrics` (Hunt, 2018) to facilitate model evaluation directly. A range of packages have also been developed specifically for model evaluation. The `hydroGOF` package (Zambrano-Bigiarini, 2017a)

provides goodness-of-fit functions specifically for comparison of simulated and observed hydrological time series. In Table 3 we present a brief overview of some other packages for hydrological statistics. For a broader overview of available packages and further examples, we point the reader to the CRAN Task View on machine learning and statistical learning (https://cran.r-project.org/web/views/MachineLearning.html).

### 3.7    Packages for static and dynamic hydrological data visualization

Data visualizations play an important role in hydrological analysis: R makes them straightforward to implement, and allows considerable flexibility. R includes three main families of graphics packages: a painter model, natively present in R and based on the S language's GR-Z model (Becker and Chambers, 1977), the Trellis graphs (e.g. the `lattice` package, Sarkar (2008)) initially implemented in S, and *The Grammar of Graphics* (Wilkinson, 1999). The last two families are both based on the `grid`

**Table 3.** Examples of packages for hydrological statistics.

| Package | Description |
|---|---|
| berryFunctions | Function Collection Related to Plotting and Hydrology (Boessenkool, 2018) |
| hydroGOF | Goodness-of-Fit Functions for Comparison of Simulated and Observed Hydrological Time Series (Zambrano-Bigiarini, 2017a) |
| hydrolinks | Hydrologic Network Linking Data and Tools (Winslow et al., In Prep) |
| hydrostats | Hydrologic Indices for Daily Time Series Data (Bond, 2018) |
| hydroTSM | Time Series Management, Analysis and Interpolation for Hydrological Modelling (Zambrano-Bigiarini, 2017b) |
| lfstat | Calculation of Low Flow Statistics for Daily Stream Flow Data (Koffler et al., 2016) |

package (R Core Team, 2018), a low-level abstraction layer that allows the development of graphical packages with different philosophies, which means that the created graphics can be stored in objects to be updated and plotted again later on.

Base-R can produce publication-quality figures; it includes a series of functions and methods that allow the user to plot various types of visualizations and the output of statistical models. Visualizations are produced in base-R using the graphics

package (R Core Team, 2018), which relies on the philosophy of superimposing elements of the graphics (with no provision for deleting an element once it is drawn), and includes a range of functions to add straight lines, arrows, axes, boxes, grids, legends, or annotation to a plot. While visualizations produced via base-R are highly customizable, other packages provide more consistent interactions and a suite of highly useful functions to summarize, highlight and split/layer data with a minimal amount of code. A community favorite is the package ggplot2, created by Wickham (2016b), which allows a high level

of flexibility and "tuning" of the graphs depending on the users' needs. ggplot2 is an implementation of the Grammar of Graphics and allows users to build almost any type of graphic (see the R graph gallery: https://www.r-graph-gallery.com). ggplot2 also contains functions like "facets" (https://plot.ly/ggplot2/facet) which allow the user to split and plot the data by categories in many different rows or columns. Figure 4 provides a hydrological example of faceting where a statistical model is visualised for different seasons (rows) and different peak-over-threshold quantities (columns).

Dynamic charts - where the user can, for instance, hover over one or multiple points to read the associated (meta)data (e.g. a hydrometric station number or the value of a point) - have also grown in popularity in hydrological analyses in recent years. These dynamic graphics are particularly useful to inspect data, such as outliers, or to explain an analysis when teaching hydrology in the classroom (e.g. by zooming in on different parts of a time series). Dynamic graphics including maps can be created by using the plotly package (Sievert, 2018), a graphic library that allows the user to make graphs interactive

with minimal extra code. A number of JavaScript-based visualization libraries can also help achieve dynamic graphics. For interactive time-series data requiring axis display features such as zooming/panning and highlighting of series/points, the dygraphs package (Vanderkam et al., 2018) provides an interface to the JavaScript charting library. The manipulate package (Allaire, 2014) allows the user to easily add sliders and other control tools to otherwise static plots. Other JavaScript-

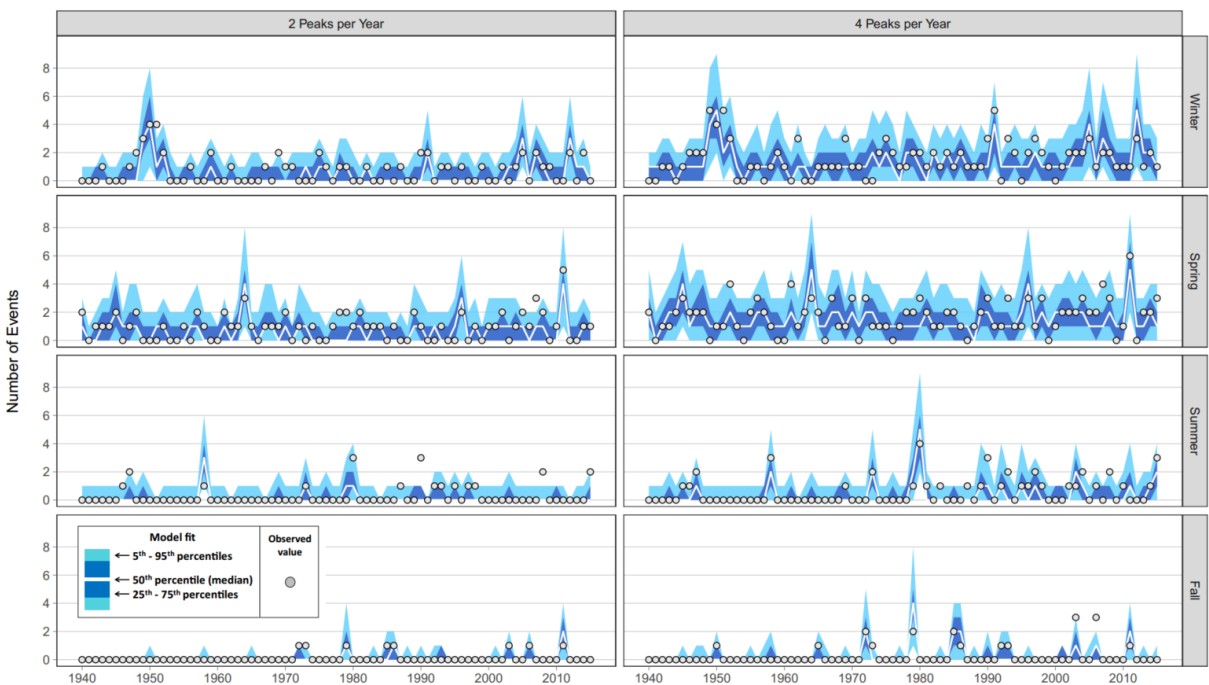

**Figure 4.** An example of a manuscript figure produced in R with the package `ggplot2` using the `facet_grid` function, which allows the user to split a dataset by categories (rows and columns). Here, a probabilistic time series GAMLSS model of flood frequency at one gauging station in Ohio, USA, is shown for four different seasons (rows), and two peak-over-threshold quantities (columns: 2-4 peaks per year). The probabilistic model is plotted in the background as color ribbons (using `geom_ribbon`) and the observed values are plotted in the foreground as grey circles (`geom_point`). Source: Neri et al. (2019).

based packages include `rbokeh` (Hafen and Continuum Analytics, Inc., 2016), which provides an interface to Bokeh, allowing the user to create web-based plots with hover and dynamic functionalities. `rAmCharts` (Thieurmel et al., 2018) creates interactive charts based on the amcharts.js library, while the `highcharter` (Kunst, 2017) package is based on the Highcharts JavaScript graphics library. Examples of all of these implementations (including `leaflet`, `dygraphs`, `plotly`, `rbokeh`,
5   `highchart`, and other visualisation libraries) can be found at https://www.htmlwidgets.org. We encourage the reader to explore these HTML widgets as they show how simple it is to implement dynamic graphics with just a line or two of R code.

Choosing the appropriate colour gradients for hydrological graphs and maps is key. It is widely accepted now that certain color schemes, and notably the infamous rainbow color scale, are poor choices for data visualization (Light and Bartlein, 2004; Stauffer et al., 2015). The rainbow scheme has been shown to distort perceptions of data, and alter meaning by creating false
10  boundaries between values, and additionally is not colourblind-safe, and other alternatives like perceptually uniform color maps have been suggested (Crameri, 2018). The R language is strong in the area of colour gradients and there are many colourblind-friendly palettes available for hydrology that follow effective data visualization guidelines (Kelleher and Wagener, 2011). There

are both manually-defined and predefined palettes using packages such as `RColorBrewer` (Neuwirth, 2014), `colorspace` (Zeileis et al., 2019), `colorRamps` (Keitt, 2012) or `viridis` (Garnier, 2018). For example, the `colorspace` package can be used for hue chroma luminance (HCL) based color palettes (which are based on how humans perceive color, in contrast to the RGB color space, which is based on technical demands of TV/computer screens) in meteorological visualizations (Stauffer et al., 2015).

## 3.8 Packages for creating presentations and documents

A vast array of packages have been developed in R for creating dynamic presentations and documents, which are particularly useful for illustrating hydrological concepts. Dynamic interfaces and web-based apps can be created with `shiny` (Chang et al., 2015), such as the `airGRteaching` (Delaigue et al., 2018a, b) interface (Figure 6, see Section 5.5). R can also be used to generate animated GIF files for presentations, for example to highlight temporal changes in land cover or reservoir levels from Earth Observation. GIFs can be created using for example `gganimate` (Pedersen, 2018) to animate ggplot2 and create videos and animated image files, or `caTools` (Tuszynski, 2018) for reading and writing GIF and ENVI binary files.

Additionally, various packages have been designed to produce interactive maps, such as `leaflet` (Cheng et al., 2018), `leafletR` (Graul, 2016), `plotGoogleMaps` (Kilibarda and Bajat, 2012), or `googleVis` (Gesmann and de Castillo, 2011). R also offers many tools for creating websites or blogposts, with packages like `rmarkdown` (Allaire et al., 2018c), `blogdown` (Xie et al., 2017) or `pkgdown` (Wickham and Hesselberth, 2018).

Presentations, books, reports and documents can be generated in LaTeX, or Markdown, natively with Sweave functionalities, or with packages such as `knitr` (Xie, 2018), `markdown` (Allaire et al., 2018a), `rmarkdown` (Allaire et al., 2018c), `pagedown` (to Paginate the HTML output of R Markdown with CSS for printing; Xie and Lesur, 2019) or `bookdown` (Xie, 2016). These packages facilitate the automation of such documents (e.g. monthly reports), through in-line text tokens that are dynamically linked to data in R's workspace.

## 4 Challenges and solutions when using R in hydrology

### 4.1 Hydrological libraries, documentation, and vignettes

For most hydrologists who are new to R, the initial hurdle is understanding how to install libraries and use packages to explore their own data sets. The book *R packages* (Wickham, 2015) is freely available online and explains everything from the basic installation of packages to the role of metadata, understanding documentation, the role of vignettes, and best practices on GitHub (one common collaboration and version control platform).

R packages centralized on CRAN are structured similarly, with a reference manual, source code, a license file, and other common elements. The code and documentation of all packages is verified before they are uploaded to CRAN. R packages ideally provide two forms of documentation: a short form (help pages) and a long form (vignettes), which are both complementary and serve different purposes (Wickham, 2015). The help pages explain what each function does, describe the required

input and the produced output, and usually include a section with executable examples. Vignettes, in contrast, are tutorials that illustrate how R packages and their functions are used, often with discussion of the outputs. However, not all packages include a link to a vignette on the CRAN repository, as this is not compulsory. Developing clear and useful vignettes is one of the key challenges in facilitating the uptake of new packages and methods; in fact these are key to reducing the misuse of software

and ensuring the package users understand the methods. Use cases can be written as blog-posts or tutorial-style papers with explanation of how to correctly interpret and implement a specific method, and helping the community move forward.

## 4.2 Integrated Development Environments (IDEs): facilitating the use of R

IDEs are software applications that are used to facilitate coding by providing the code editor, compiler or interpreter and debugger within a single graphical user interface (GUI). There exist a range of IDEs for R, such as Eclipse/StatET, (an Eclipse

based IDE; e.g. des Rivières and Wiegand, 2004; Wahlbrink, 2016), Emacs Speaks Statistics (ESS, an add-on package for GNU Emacs; Rossini et al., 2004), Microsoft's R Tools for Visual Studio (RTVS, a plug-in for the Microsoft Visual Studio IDE; e.g. Beard, 2016), RKWard (which integrates with the KDE desktop environment; Rödiger et al., 2012), Tinn-R (a replacement for the basic code editor designed to replace Rgui; Faria et al., 2008), or Vim/Neovim (a terminal-based working environment designed to facilitate working in a pure command-line interface; e.g. Neil, 2018).

The RStudio IDE (RStudio Team, 2018) is the most popular of the IDEs. RStudio facilitates the uptake of R by hydrologists, by providing a helpful research tool and training environment to new and experienced programmers alike. RStudio is available in two editions: an open-source edition and an enterprise edition with a commercial license. Both can run on a desktop (RStudio Desktop) or server (RStudio Server) and can be installed on different platforms (Windows, macOS, and Linux). Most hydrologists use the free RStudio Desktop edition, although university departments and companies increasingly offer server

editions too. RStudio's features include: a console, a syntax-highlighting editor that supports direct code execution, integrated R help and documentation, support for version control systems, as well as tools for plotting, history, debugging and workspace management.

The RStudio environment makes it straightforward for hydrologists to conduct a range of tasks, e.g. visualise data, create dynamic graphs or web-apps with `shiny`, use version control, or develop and view Rmarkdown presentations e.g. in Latex,

HTML or PowerPoint), all within the IDE. Other advantages of RStudio include the easy access to the help window, the ease of package management and updating, code debugging, or even the easy development of packages in one location. All of these benefits contribute to facilitating open hydrological science. RStudio version 1.2 includes many new features to help interact with other programming platforms, such as the `reticulate` (Allaire et al., 2018b) and `r2d3` (Luraschi and Allaire, 2018) packages, which make it easy to call Python and D3 from within R sessions and R Notebooks. An exhaustive list of features

can be found on the RStudio website (https://www.rstudio.com/products/RStudio).

## 4.3 Big Data and parallel computing challenges in hydrology

In the early years of R, the software was unable to handle large data files exceeding millions of rows or complex data formats. However, both of these limitations have since been overcome. Some of the early packages for handling large data files include

bigmemory (Kane et al., 2013) or biganalytics (Emerson and Kane, 2016), where matrices are allocated to shared memory and use memory-mapped files. More recently, the package data.table (Dowle and Srinivasan, 2018) has gained increasing popularity for enhanced data frames, allowing fast aggregation of large data (e.g. 100GB in RAM) or fast ordered joins. In hydrology, some of the spatial data handling packages are particularly relevant, such as the sf package (Pebesma, 2018a) for large shapefiles, or the feather (Wickham, 2016a) package for feather files, a lightweight binary columnar data store designed for maximum speed. Database connections can be established using a range of packages such as RPostgreSQL (Conway et al., 2017) or RSQLite (Müller et al., 2018). Distributed dataframes can be implemented using packages such as sparklyr (Luraschi et al., 2018).

As the volume of available data increases and hydrologists use a greater number of models and ensembles, parallel computing - where many calculations are carried out simultaneously, instead of sequentially - has become an essential tool in computational hydrology and has sped up analyses. For instance, instead of using traditional for-loops (an approach were one action is carried out over a data set iteratively), the data may be broken down into groups (e.g. by year or by season) and functions can be applied to each group in parallel. The performance boosts that can be achieved by parallelizing the code (i.e. using more than one core at a time) are considerable. Even without access to a high-performance computer or cluster, it is possible to perform hydrological tasks faster, since most local machines now have between four to 16 cores. R has multiple facilities and packages to enable the parallelization of code execution. At the most simple level, base-R functions like lapply and sapply can be used to apply a specific function to a vector/list input, which can speed up analyses considerably. For instance, the base-R parallel package for distributed computing provides methods to access additional computational power by allocating available processing cores into a cluster. The foreach (Microsoft and Weston, 2017) and doParallel (Microsoft Corporation and Weston, 2018) packages can also be used together to execute "for-each" loops in parallel.

Other packages that are widely used for parallel computing in hydrology and other areas include the snowfall (Knaus, 2015) package, which facilitates the development of parallel R programs, e.g. by including extended error checks. Additionally, some packages have parallel functionalities that are integrated directly within the package, such as the h2o package (LeDell et al., 2018) that provides an interface to H2O, a scalable platform offering parallelized implementations of supervised and unsupervised machine learning algorithms.

## 5 A roadmap for the future of R in hydrology

As we have shown above, the development of R fosters progress in hydrology. In this section we discuss what we perceive as some of the future avenues for enhancing hydrological research or operational hydrological practice with R, and how we can achieve these as a community.

### 5.1 Sharing R code with the community

Open research practices bring significant benefits to researchers (McKiernan et al., 2016), such as increases in citations, media attention, potential collaborators, job opportunities and funding opportunities. Most importantly, sharing code increases the

likelihood that an approach will be used by other scientists in their research, and saves new users the trouble of "reinventing the wheel" and writing codes that have already been developed by others. There are many different platforms for sharing and publishing R code such as GitHub, Figshare, and RPubs/Plotly (also for dashboards and interactive plots). To obtain a DOI, users may wish to use repositories such as ZENODO, a general purpose open-access repository which allows researchers in the sciences and humanities to deposit data sets, research software, and reports. The data publisher PANGAEA, additionally, is specifically tailored to archiving, publishing and re-usage of data in Earth and Environmental sciences. For further guidance, some of these platforms are discussed in the code and data policy section of the open-access journal *Geoscientific Model Development*.

## 5.2 R packages as a driver of progress in hydrology

The consistent stream of new R packages has been a great driver for progress not only in hydrology, but even in science more broadly, as packages favor the uptake and development of methods. Additionally, the open-source nature of R packages means that different users can contribute feedback to R package developers and help enhance existing code. R users can raise issues for certain packages directly on online platforms hosting the repositories (such as GitHub or Gitlab) to foster an online-documented discussion with the package developers (generating interaction in the community). Package authors can also add a bugs report link or an email address in their package Description file (a common best-practice for developers) to specify how they prefer to receive bug reports. This feedback between users and developers is one key route to scientific progress. Users can identify issues or suggest improvements by commenting on online collaboration platforms (e.g. GitHub or GitLab) or by emailing the developer/maintainer. Most packages are hosted on these repository hosting platforms before becoming available via the CRAN archive, ensuring a certain standard and best practices are met. Although CRAN itself does not have any mechanism to check the quality or cleanness of the code, there is a suite of packages that are used to (i) ensure clean code and documentation that follow a widely accepted style-guide, such as the `lintr` (Hester, 2018), `goodpractice` (Csardi and Frick, 2018) and hunspell (Ooms, 2018) packages, and (ii) rigorously test the functions against pre-defined outcomes, using for example the `testthat` package (Wickham, 2011). In comparison with GitHub, CRAN is a specific organised repository which provides executables for several platforms, whereas finding an hydrological R package on Github can be difficult. The comparison between the two is somewhat like visiting a curated art auction versus an antique store. Additionally, CRAN has more thorough checks and gives visibility/searchability to a package, also signalling that the author intends to maintain the package over the long term.

Developing an R package requires a structured approach, just like writing a scientific paper. There are many generalist resources for writing R packages, such as the R packages book (see http://r-pkgs.had.co.nz, Wickham, 2015), or a suite of video tutorials on YouTube. The main requirements for publishing a package are listed on the CRAN website. Providing documentation and tutorials, e.g. via blogposts or good Readme files, is essential. Package authors can also provide user-friendly and modular functions to aid maintenance and future development (e.g. through external contributions or suggested changes on online platforms). The RHub service also provides a "build and check" service for R packages (Csárdi, 2017).

Authoring hydrology-based R packages that can stand up to scientific scrutiny and ensure user-friendliness is not a minor task, and such investment should be recognized within the community. Fueling the development and dissemination of new R-based methods is therefore the joint responsibility of developers, authors and journal editors. If developers include digital object identifiers (DOI) as well as instructions for citation within their software, then authors can subsequently cite these packages. The adequate reference(s) can be obtained with the citation function for every package. The references generally comprise a reference to the package (with CRAN link and package version, which is key for encouraging reproducibility) and sometimes also include an additional journal paper reference. If both are available, then ideally both must be cited to provide recognition of computational scientists' contribution to the hydrological community and to enhance the reproducibility of the research. Editors might also support reproducibility via special issues or sections for technical notes. Dedicated software journals such as the *Journal of Open Source Software* can be used to publish brief, technical descriptions of R packages. Any potential apprehensions for publishing such methods (e.g. due to a lack of scientific scrutiny) can be alleviated through software peer review initiatives such as those provided by rOpenSci. It is worth noting that when writing new packages, the open-source approach has both pitfalls (risk of errors in new packages) and strengths (community review, and range of tests that can be implemented via the above-mentioned packages).

## 5.3 Harmonizing hydrological workflows

One of the main challenges today for hydrological workflow harmonization is to standardise calls to data which are structured in different ways by data providers around the world. While the number of packages for hydrological data retrieval has grown considerably (see Table 1), the packages are structured differently because they were set up independently and because the underlying hydrological and hydrometric datasets differ from country to country. As more nations develop similar hydrological data acquisition packages, we believe it would be worth implementing a common syntax and data output form. For example, it would be ideal to use consistent APIs and output objects across packages. Suggesting the "ideal" format is beyond the scope of this manuscript, but a potential task force might be set up to help draft a way forward for the community. There is currently no effort to combine data retrieval packages from different regions, but it might be worth implementing a meta-package with functions that convert hydrometric data from other packages to a standard format, or one for hydrological models to be run within the same framework. Additionally, the growing tendency towards data standardization at the global level may facilitate the development of consistent data retrieval packages. Examples of global data standards include WaterML, the Open Geospatial Consortium standards for hydrological time series, or TimeSeriesML, a more generic candidate standard currently under discussion. As more water agencies and data providers adopt such standards, it will become much easier to develop consistent data retrieval packages, and in theory we may no longer require different packages for data retrieval because the only component that should change is the server endpoint.

## 5.4 APIs: hydrological data acquisition and provision

Application programming interfaces (APIs) play a key role in hydrological data acquisition and provision, and are likely to become increasingly important in the future. An API is a set of code which usually includes subroutine definitions, communi-

cation protocols, and tools for building software and interacting with different datasets. APIs are use-case specific by definition, but the interface is often provided in HTTP (i.e., Web protocol), so requests and responses can be made and received by a wide range of languages or systems. For R-Hydro users, hydrological interaction with HTTP APIs usually comes in the form of data acquisition packages such as the aforementioned packages `rnrfa` (Vitolo et al., 2016a), `tidyhydat` (Albers, 2017), or `daymetr` (Hufkens et al., 2018). These packages provide means of posting requests via the R command line, and then return an R object for subsequent manipulation. Internally, the source code translates all inputs and outputs to and from HTTP. Programmatically accessing a variety of vast online resources (i.e. data) in this way has considerably advanced our understanding in the earth sciences.

Recent developments, however, have opened up a new, and arguably under-utilized, approach to APIs for the R community: rather than exploiting an existing interface, R users can now increasingly rely on a set of tools to develop and make accessible their own APIs for use by third parties via HTTP. Noteworthy projects here are OpenCPU (client and scalable cloud implementation, Ooms, 2014) and RStudio's `plumbr` package (Lawrence and Wickham, 2018). API development is likely to become a major avenue of future advances in scientific computing within the hydrological community, as well as interaction beyond the scientific realm. Such APIs can provide access to latest raw or analyses-ready data, methods, or even entire (statistical) models written in R.

The simplicity and ubiquitous implementation of HTTP has vast implications, of which we highlight four. 1) Common issues with interoperability between languages can be overcome, and more attention can be afforded to gaining insights, rather than developing (often convoluted) language bridges (termed a "separation of concerns"; Council and Heineman, 2001). This means, for the first time, fully seamless interaction between research groups working in different languages is possible (e.g. between R, Python and MatLab). 2) Computational infrastructure can be scaled with comparatively little technical knowledge, increasing R's aptitude to a wider range of applications. 3) Outputs can be made accessible to third parties with or without a deeper understanding of the underlying science. 4) Empirical data is rarely "clean" and scientific computing hence has to be adaptable: R source code can be altered to account for changing data, while maintaining a standardized, public-facing API. To highlight potential strengths of such APIs, we provide a brief, interdisciplinary use-case scenario rooted in natural hazard management and hydrological sciences in Figure 5.

## 5.5 Teaching hydrology in R

Due to its relative ease of use and open-source nature, R is increasingly being used as an interactive tool for teaching in the hydrological sciences. Many examples of typical hydrological analyses can be found online as tutorials (see for example, Hurley, 2018). The USGS have also published relevant examples on their hydrological blog at https://owi.usgs.gov/blog which may be of interest to the community.

R packages such as `airGRteaching` or `TUWmodel` have also recently been developed to facilitate teaching of hydrological modelling. The `TUWmodel` package (Viglione and Parajka, 2014) provides a lumped conceptual rainfall-runoff model, based on the well-known HBV model (Bergström and Forsman, 1973) and is designed for education purposes. The `airGRteaching` package (Delaigue et al., 2018a, b) provides access to a suite of GR rainfall-runoff models such as GR4J

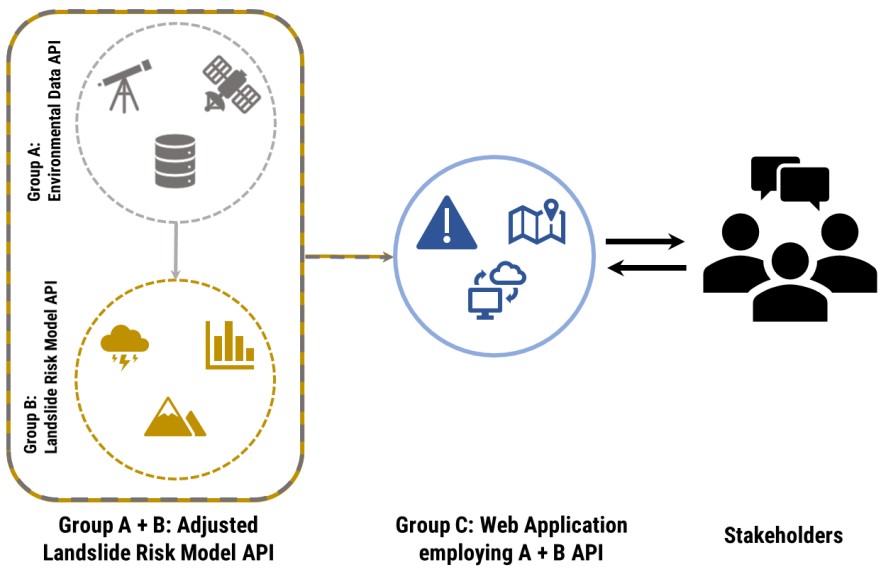

**Group A + B: Adjusted Landslide Risk Model API**  **Group C: Web Application employing A + B API**  **Stakeholders**

**Figure 5.** Use-case scenario for implementation of APIs developed within hydrological sciences. Group A collates the most recent earth and ground-based observations of land cover, topography and climate data for an in-house project, yet provides an API to interact with an analyses-ready, spatially-explicit data set for their region of interest. Group B is interested in geomorphological processes - and has developed a model to predict likely occurrence of mass movements, which they also provide access to via an API. During a period of extreme rainfall over a scientifically unrelated region, both groups decide to adjust their methods and API to provide pertinent data (group A), which feed into predictions of landslide risk (group B). Finally, a disaster relief organization (group C) can swiftly act and use the outputs from the adapted API (A + B) to develop a simple web application with maps and warnings for use by the general public in the affected area.

for use by students with limited programming skills: only three basic functions are needed to prepare the data, calibrate a model and run/evaluate a simulation, and a single function can produce plots directly using objects from the three previously-mentioned functions. In addition, a graphical `shiny` web application is included to explore data and model results with automatic updating of the plots and scores when the model parameters are modified by students; the app allows users to ex-
5  plore the internal fluxes and state variables of the models (Figure 6). Such tools can be used for understanding and illustrating the main hydrological processes and concepts and for relating them to hydrological models' components and parameters.

There is now an increasing number of online applications that allow beginners to learn R in a sandbox, i.e., a virtual space for testing coding online. Sandboxes are particularly useful for introducing basic methods in computational hydrology, without having to master the technicalities of R. Examples of R sandboxes include the RStudio Cloud (https://rstudio.cloud), which can
10  be used to learn, teach, or test R code. Another sandbox, initially called R-fiddle, is now called datacamp-light (https://github.com/datacamp/datacamp-light). Many other R sandboxes can be found online. Thus, rather than installing R on multiple PCs, sandboxes can be used by educators/lecturers by asking students to connect directly to the web.

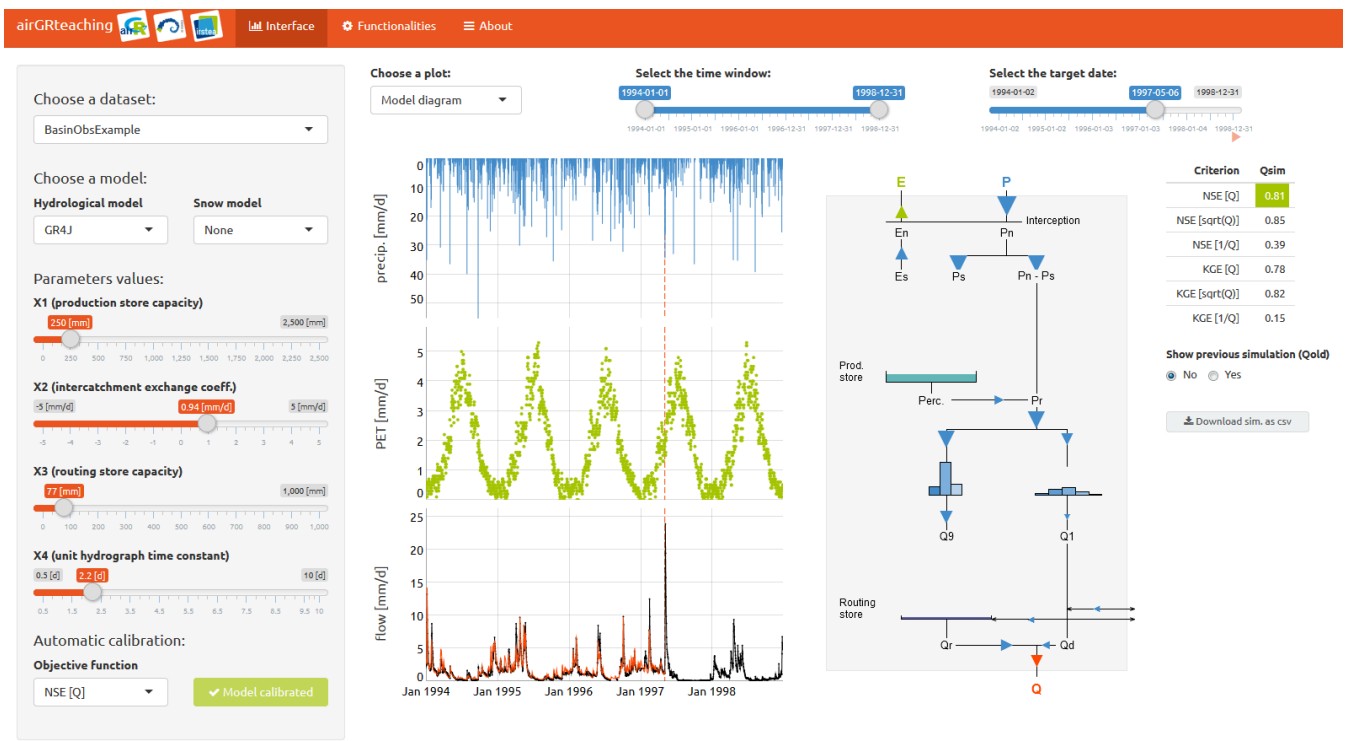

**Figure 6.** The `airGRteaching` interface that is used for teaching hydrological modelling. Students can choose a catchment dataset, a hydrological model and activate a snow model on the top left panel of the interface. Modifying the parameters values (on the left) by using the sliding bars will automatically update the graphs and scores displayed (on the right). This interface highlights in an interactive manner the basic principle of the rainfall-runoff relationship and its description in hydrological models. On the bottom left of the interface, an objective function can be chosen and an automatic calibration procedure can be launched within the interface.

## 5.6 Developing the community: short courses, help desks and meet-up events

This paper reflects the strong collective desire to develop the community of computational hydrologists in R. As mentioned in section 2.5, the R-Hydro community has been meeting regularly in recent years at the EGU General Assembly during the Short Course "Using R in Hydrology". This course is run annually by the Young Hydrologic Society (YHS) and is typically
5   attended by a wide range of hydrologists, ranging from beginners to more experienced users (Figure 2). The resources and teaching presentations from the short course are made available to the community online on the YHS GitHub pages (https://github.com/hydrosoc/) and archived on the Zenodo repository (see Boessenkool et al., 2017; Slater et al., 2018; Hurley et al., 2019). These presentations include a range of topics such as: obtaining, cleaning and visualizing hydrological data with R; parallel and high performance computing for hydrologists; automating tasks; using R Shiny with hydrological data; modelling
10   the hydrological cycle in snow-dominated catchments; community-led initiatives; using R as a GIS; discharge time-series

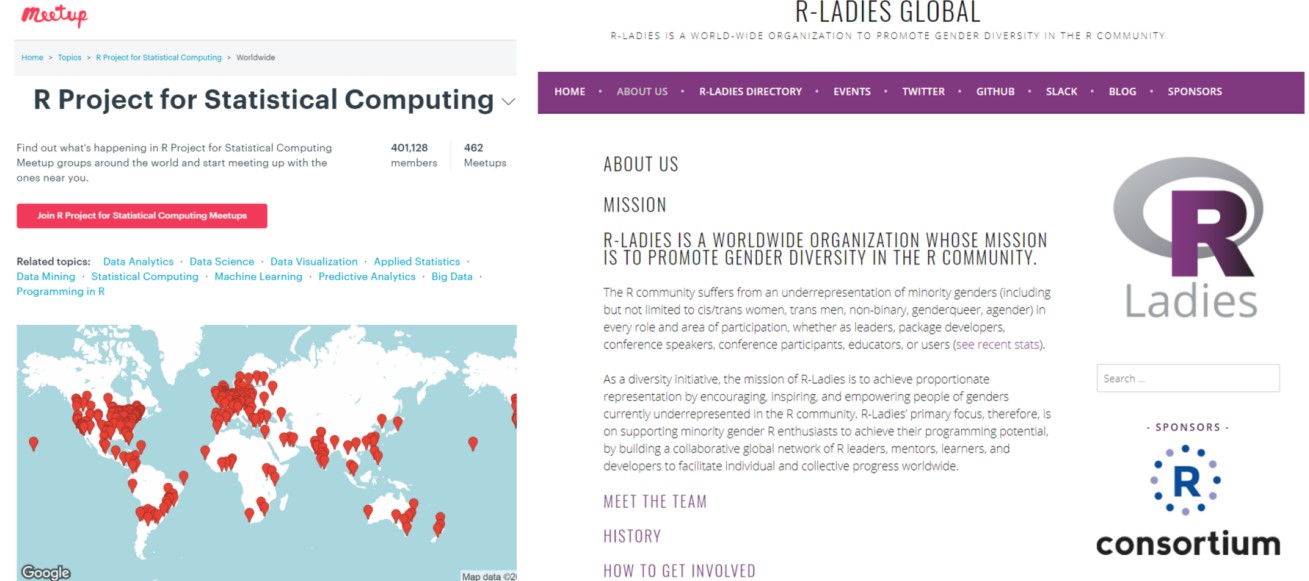

**Figure 7.** The R community. Left: the R meetup groups https://www.meetup.com/topics/r-project-for-statistical-computing; Right: the R-Ladies global organization https://rladies.org/about-us.

visualization; extreme value statistics; hydrological modelling; trend analysis; accessing hydrological data using web APIs; extracting gridded netCDF data; report generation; and other hydrological tasks in R.

Many other R conferences will be of relevance to the readers, including the *useR!* conference, *RStudio*, *satuRdays*, or *eRum*. These conferences are not specific to hydrology and earth sciences, but usually have some discipline-specific sessions as well

as sessions and short courses of general interest for learning R, including best practice, spatial analysis, statistical methods. Many of these conferences also stream the talks and/or make resources available after the event.

The global reach of R meet-up groups has also grown rapidly, such as the official R-users group meet-up (http://r-users-group. meetup.com). Many local initiatives for R users have regular meet-ups and seminar series and can be found either on the meetup website or on social media webpages, such as Twitter (e.g. https://twitter.com/rusersoxford). There is also an ongoing push for

greater inclusivity of under-represented groups in programming, as exemplified by the rapid growth of the global "RLadies" movement (Figure 7).

In addition, the computational community has been trying to provide support to other programmers by running "Coding Help Desks" such as at the American Geophysical Union (AGU) 2018 Fall meeting, in the Career Center. Geoscientist volunteers ran the desk in 2018 to provide perspective and advice to other coders, with a range of short, 10-minute tutorials on topics such

as keeping track of code versions with GitHub, making high-quality plots, or using a new package/library. We anticipate that such help desks, short courses and meet-up sessions will continue to help grow the computational hydrology community in future years.

# 6 Conclusions

Over the last decade, the open-source programming language R has acquired a central role in hydrological research as well as in the operational practice of hydrology. With the rapidly increasing number of packages that are now available for every step of the hydrological workflow, R facilitates a broad range of hydrological analyses from start to finish. This paper provides an overview of the use of the open-source programming language R in hydrology, by describing these packages as well as the influence of R on the discipline. Both the flexible nature of the language and the diverse range of computational, visualization, and modelling tools (physically-based and statistical) have facilitated the testing of hydrological theories over a range of spatial and temporal scales, as well as interactive teaching of hydrology within the classroom.

By encouraging others to use the language, to share their codes, propose new packages or contribute to the improvement of existing packages, we believe R will continue to facilitate further advances in hydrology, with wide-ranging improvements of hydrological theory, models and tools. These new computational tools and approaches are essential to achieving long-term goals such as the IAHS's Science Plan for the decade 2013-2022, "*Panta Rhei*: Change in Hydrology and Society", which seeks to improve the assessment, attribution, and modelling of hydrological change. The rise of computational hydrology is also playing a key role in enhancing the reproducibility of science and the computational literacy of both scientists and practitioners. Within scientific research labs, we anticipate that committing code to a repository will become standard practice, as will the submission and review of code along with manuscript text as part of the scientific publication process. Ten years from now, with the continued rise in the teaching of open-source programming in schools - and of computational modelling within university curricula - it is plausible to expect that R will play an increased role in hydrology.

*Competing interests.* The authors declare that they have no conflict of interest

*Acknowledgements.* We thank two anonymous reviewers and Michael Stoelzle for comments that improved the manuscript. We also thank all those who have contributed to the R-Hydro community and the R community more broadly, whether by developing and sharing their code, or helping and teaching others to use these tools. In particular, we thank the guest speakers who have participated in the "Using R in Hydrology" EGU short course. Finally, we also thank Paul Astagneau, MSc at Irstea, for the discussions on modelling packages that helped improve the manuscript.

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
