# Peer review of "Using R in hydrology: a review of recent developments and future directions"

_Hydrology and Earth System Sciences, 2019_

## Referee Comment (RC1) · Anonymous Referee #1 · 3 Mar 2019

This manuscript represents a substantial contribution to scientific progress in hydrology in that it makes readers aware of the new task view in R for Hydrological Data and Modeling. Because there are almost 14,000 packages available in R and countless more in other repositories, it is difficult for analysts to keep up on packages potentially useful to them. Having a hydrology Task View is an important maturation in the use of R in the hydrology community, that many might not be aware of for quite some time without some targeted messaging to the hydrology community.

The discussion of many packages available to hydrologists is good. While the authors cannot discuss them all, they have a fitting selection of useful packages. For example, an analyst may be aware of the trend analysis functions in R in multiple packages, but not the spatial analysis/GIS functionality, and this article helps brings together diverse

analysis methods in one discussion of the hydrology community.

The mention of the traceability of package authors through their ORCIDs is another important, relatively recent addition to R. As an active member of the R community, I was unaware of this and found that many packages do not use this yet. However, this is another way to judge the background of package developers and find related articles—excellent information to provide to the R-Hydro community.

The scientific quality of this article is excellent, providing a brief history of software languages used in hydrology and a brief history of R, while making good arguments for the use of R in hydrology. Reproducible research benefits all scientific work and is an area with continued need for improvement in hydrology—R can help.

The figures and graphics are appropriate for the article and are informative. The paper is well written and carefully edited prior to review (much appreciated by reviewers).

The scientific resources listed were good. It could have even been better if some of the authors were from the US and able to describe additional developments, providing a more global view.

A suggested improvement to the article is the inclusion of the useR! Conference, https://www.r-project.org/conferences/ in the discussion of scientific resources and courses. These annual conferences are not specific to hydrology and earth sciences, but usually have some environmental science or ecology sessions as well as sessions or short courses of general interest on learning R, best practices, spatial analysis, and statistical methodology incorporated in various R packages.

Another suggested improvement is more discussion on setting up repositories. What are a few good options for hydrologists? Many subscribe to the workflow provided, but are still somewhat mystified by the repository step, that is actually rather easy once someone clears the initial hurdles of selection and setup.

On page 14, lines 8 and 9, the "(Mann-)Kendall testing" punctuation is a bit awkward,

exacerbated by the line break in it. I suggest leaving it as Mann-Kendall testing because readers will know what that is referring to whether they know it as Kendall or Mann-Kendall.

---

## Referee Comment (RC2) · Anonymous Referee #2 · 4 Mar 2019

I enjoyed reading this paper, which is for sure useful to get an overview on what R can do for hydrologists and what hydrologists can do for R. I am an R user, and I have also developed packages, which places me at one side of the spectrum of potential readers of this paper. As such, I can benefit a lot from the information provided, especially regarding tools on subjects I am not an expert in. I wonder whether the great amount of information provided in the paper isn't overwhelming for a R beginner. It is unlikely that a R beginner will comment the paper in this discussion stage so, if I were one of the Authors, I would suggest to ask, e.g., a couple of master or starting PhD students to perform an internal review of the paper as non-expert readers (that's just an idea).

Besides the many advantages of using R (and open data and software in general), I would have expected a mention on possible drawbacks. A couple of examples follow:

[Figure]

- The easy accessibility and usability of open tools results, sometimes, in applications without critical scrutiny on what one does. It happened to me to receive an email from a user of my package asking me to tell her/him how to apply the methods to her/his data. A few months later I've been asked to review the paper written by the same R user who, then I realised, had no clue about the meaning of the methods applied. In other words, how can the misuse of software be reduced within the open source community? (I am aware that this happens also with proprietary software).

- Everybody can make an R package available, even if not carefully tested. This may result in errors that are then propagated by users. That happened to me too. I had a bug in my code but, thanks to the open source, a user spotted the error and I could correct it. When my package was first published, I received several feedback messages that helped me a lot to clean the code. The frequency has then decreased considerably meaning that, hopefully, the main errors have been solved (or that people do not use the package any more :-)

Having said this, I believe the paper is good and worth publishing in HESS, maybe after a minor revision.

Small things:

Page 2, line 31: I would mention here more specifically what R is, or better what R was, i.e., software for statistics (you say that in section 2.3). This is to better inform the non-expert reader.

Figure 1: I am not a big fan of this kind of figures in review papers. The text describing it could be sufficient.

Page 5, line 18: I would say here that hydrologic research is criticized for the lack of reproducibility, rather than scientific research in general.

Page 7, line 14: Maybe it is just the way I call things, but I would say that C and Fortran are very efficient in performing loop tasks. R, Matlab etc. are very good in vector

operations.

Section 2.5: Just as a note, which may be ignored by the authors if not relevant, IAHS also organises R courses in its meetings, e.g., there will be a "Using R in Hydrology" workshop at the IUGG2019 conference in Montreal. Also, among others, the PUB summer course https://www.waterresources.at/index.php?id=5&tx_ttnews%5Btt_news%5D=42&cHash= 1f78fb6f0a0bf08ca5593890cc294f84 uses R intensively.

Page 10, line 16: I would not use Twitter to discover packages, but to be exposed to communications about new packages or new features.

Page 12, line 24: I normally use RgoogleMaps https://cran.r-project.org/web/packages/RgoogleMaps/index.html to interact with Google Maps.

Page 13, line 15: Actually also TUWmodel can be used in a semi-distributed fashion, allowing for differentiating into elevation zones. Page 14, line 19: For regional frequency analysis I would also mention the lmomRFA package https://cran.r-project.org/web/packages/lmomRFA/index.html developed by the very father of the L-moment methodology J. R. M. Hosking.

---

## Referee Comment (RC3) · Michael Stoelzle (Referee) · 2 Apr 2019

First of all, I really support the idea of the paper and I think the presented review is fully in the scope of HESS. I like how the relationship between hydrology and R is presented and the authors gave a broad overview of historical and recent developments in the R-Hydro community and sketch a interesting roadmap for the future of R in hydrology. The paper is from my perspective also a good example for a joint effort from young scientists in our hydrological community. However, since the first two reviews are full of praise I try to be a little bit more critical putting together some thoughts that hopefully improve the paper. My major "overall" suggestion is to revise the paper going a little bit away from just having a list of available packages or presenting potential possibilities in the R-Hydro universe towards a) more "best practices" for certain challenges and b)

[Figure]

more comparisons of similar packages. However, I absolutely recommend to publish this paper in HESS. See more specific comments below.

Best regards, Michael

Major comments

**The authors give a lot of technical information about R in different section in the paper (e.g. GNU S background in sect 2.3, R script workflow in Sect. 2.1, R output format in the Introduction). As all the parts are really important to learn about the fundamental principles of R for me it would be more helpful to have the technical and historical information about R in one separate section. This might be also helpful for readers who are new to R coming from other programming languages.**

**I miss more discussion about different philosophies how to program in R. From my perspective there is, for example, often the question whether you are doing your visualizations with ggplot2 or in base-R or doing data wrangling with dplyr or in a complete different "style" with data.table. I don't know if there are comparable examples for hydrological packages (doing more or less the same thing but with different techniques / packages). The paper could, however, be improved by presenting opposed approaches to solve problems in R. For example, data.table is known to be very fast in data wrangling but on the other side the dplyr package (and piping %>%) is known to support highly readable code. Here the trade-off between performance and communication could be discussed (i.e. Sect 4.3 and Sect. 2.1/2.2). My experience from teaching R in hydrology is that students really like the concept of piping (%>% read as "and then do that") and can do the step from base-R to tidyverse in a couple of sessions. Tidyverse and Pipes could also be mentioned in Sect. 5.3 as a good possibility to teach "highly readable code" in the classroom.**

**Regarding Sect 2.2 it would be helpful to draft a way forward how developers of R-Hydro packages can develop better R packages with a clear description of input data types, a comprehensive documentation on package usage. For example, tidyverse is**

set of packages that work well together as all packages use the same data representations and API design. Is this also a valuable approach for, let's say, developer of R packages that include hydrological models? For example, is it possible to use the data retrieval packages from Table 1 in a coherent way or is the syntax between all packages different? Is there an effort to combine data retrieval packages from different regions with a consistent R syntax/usage? Using different hydrological R packages with the same syntax might be of great value for the community. A short discussion about this issue can draft a way forward in the R Hydrology community (as mentioned in Sect 4.1 where the need of good vignettes in R is highlighted).

**The introduction of section 3 and also Fig. 3 are really nice and well-conceived. I like the idea to split the data analysis into different parts and subsections in the paper (Sect. 3.x).**

**In general, the paper is full of valuable package, links, and corresponding citations. However, from my perspective often only the package names are given and more detailed information how the package is working or why it is especially useful/important for the hydrological community is missing. Of course, it is not possible to describe all the functionality of all the packages in detail but the authors could pick 2-3 cases where they really compare the advantages and disadvantages of using different packages for the same purpose / challenge. To be honest (and I also think that the author team is aware of that), you can relatively easy check on the internet if there is a package for a certain task and what are the technical details or data requirements of a single package. So, the great list of available packages cannot be the main message of this compelling paper! Comparing different packages among each other could really be of great value for the hydrological community and this paper is a good place to do that. A good example is Sect 3.4 describing spatial data analysis and visualization in R. In this section a lot of important R packages are mentioned but the reader gets less information which package is better or has the same functionality as other packages etc. In Sect 3.5 the information about snow functionality in the hydrological models is a good**

example how this can be done.

**For me a major deficiency of the paper is that the topic "colours" is completely missing. Hydrology is a multi-facetted discipline and hydrologists doing a lot of visualizations for posters, presentations and papers. Hydrology is also often based on multi-dimensional analyses with different variables in space and time. To visualize hydrological processes and hydrological change colour is often the first choice to compare data sets and to investigate relationships. As we all know, a sophisticated colour choice is often a huge challenge. I encourage the authors to implement a "colour section" where a short discussion about, for example, the basics of appropriate colour gradients for graphs and maps is given (one-colour gradient, two-colours gradient, discrete vs. continuous variables). In large-scale hydrology "rainbow" colour gradients are still state-of-the-art (to generate fancy and colorful world maps) but a lot of papers and blogpost have taught us that the "rainbow" isn't perceptual uniform and not colourblind-safe.**

Minor comments

P2L20 Give 1-2 examples of hydrological models coded in R.

P2L33-35 It might be helpful for the reader to compare the fundamental principles of R against other programming languages (e.g C, Java) to better describe the architure of R (e.g. in terms of object-orientantion, complilation of code,

P3L4 Are there any reasons for the large number of updates in R-Hydro packages (related to Fig.1)?

P4L11 The platform Stackoverflow (from Sect. 2.5) can also be mentioned here as there often also information on relevant R packages are given.

P5L31 What is meant with the "standardized format"? Here it might be also valuable to add packages like roxygen2 that help to develop, create and document R packages.

P6L27 It will be easy to find some justification for this statement (e.g. number of R

topics on stackoverflow.com compared to other programming languages).

P6L3-14 The paper would be improved by adding various possibilities to share and publish code during and after paper publication (e.g. github repository, gists or other tools and platforms). Is there a possibility to share R code with a DOI?

Sect 2.3 For me this section contributes less to the strength of the paper – what is the key message here? The specific statistical packages that are relevant for R-Hydro are given in Sect 3.6 (as mentioned). The link to CRAN is perhaps too unspecific here. Yes, R provides a vast number of statistical tools for hydrologists but then you have to explain in more detail why and what advantages R here has to offer (compared to other languages?).

Sect 2.5 What are the rotating topics and why have they been chosen? The picture might be nice to see but the space could instead better be used to highlight the EHGU SC topics from the last years with a short information why the topics were of great interest for the community. It might be also important to highlight the possibility to raise issues for certain R packages on github.com to foster an online-documented discussion with the package developers (i.e. interaction in the community). At least a cross reference to Sect. 5.4 could be made here.

Table 1 I am not sure on which basis you have choose the 11 packages? Are they the most important ones? Perhaps a thematic order would improve the table (sort by data type, spatial extend, multiple proposes or not,…)

P11L5-9 Consider to mention the rio package here as it is really powerful. The tidyverse package is cited as a package that reads multiple file formats – I think this is not the main purpose of the tidyverse package.

P11L10 Which netcdf package is the best? Or let's say, are there more information which package to use for a specific application.

P12L15 For all ggplot2 users it is really important to mention that the ggplot2 3.0 version offers support to visualize sf objects directly with a specific geom (geom_sf). So, if one is familiar with ggplot2 and want to visualize spatial data (raster, shapefiles etc.) this is a really easy way to do that? As the ggplot2 community is growing and growing this might be a really important information for some readers of the paper.

Table 2 Should be included in Sect 3.5

Sect. 3.6 Package "skimr" might be valuable for this section.

P15L10 I agree, one strength of ggplot2 and faceting is the ability to generate multiple views. This is especially important to avoid overplotting or – as the authors mentioned – to split a data set in different plots using meta information of this data set (i.e. season). If you have a nice example for faceting here would be a good place to show this principle for the readers.

P19L17 Is it true that publishing a R package on CRAN is free of quality checks? However, a short comment on the differences during package publication (CRAN vs. GitHub) might be helpful here.

P19L21-26 I agree but the paragraph could be more specific on how we should develop R packages. Is there a good article or blogpost describing the progress of publishing a R package? What are the requirements to generate good tutorials or readme files? Might it be useful if journals like HESS recommend to publish paper code on github?

---

## Author Response (AR1)

*Response to Reviewers*
**"Using R in hydrology: a review of recent developments and future directions"**

Previous responses from the online discussion are in blue; new responses after revision are in red.

**Response to Editor**

5   Editor Decision: Publish subject to revisions (further review by editor and referees) (24 Apr 2019) by Erwin Zehe

*Comments to the Author:*

*Dear Dr. Slater,*

*After reading the reviews and your manuscript again, I share the generally very positive attitude of the reviewers. Overall I agree with Michael Stoelzle that the manuscript would gain even more strength, when putting more emphasis on discussing*

10   *best practice of using R code in hydrological research. This will be particularly valuable for R beginners.*

*Best regards,*

*Erwin Zehe*

We would like to thank the Editor for his positive comments on our manuscript. We have incorporated the vast majority of the Reviewers' suggestions. Throughout the manuscript, we have emphasized our discussion of best practice in using R code

15   for hydrological research. For example, we have discussed: setting up repositories and committing code online; following a standard "hydrological workflow" in R; sharing R code via the "literate programming paradigm", blog posts, tutorials; and designing hydrological vignettes to facilitate the uptake of R packages.

**Response to Reviewer 1**

We would like to thank Reviewer 1 for constructive comments on our manuscript. We feel that many of these comments are helpful and will improve the quality of our review paper. Below we provide the Reviewer's comments verbatim in black italic text, and our responses are immediately below each comment in blue text.

25   *"This manuscript represents a substantial contribution to scientific progress in hydrology in that it makes readers aware of the new task view in R for Hydrological Data and Modeling. Because there are almost 14,000 packages available in R and countless more in other repositories, it is difficult for analysts to keep up on packages potentially useful to them. Having a*

*hydrology Task View is an important maturation in the use of R in the hydrology community, that many might not be aware of for quite some time without some targeted messaging to the hydrology community"*

We thank the reviewer for the positive comments on the manuscript, and we agree that it was important to highlight the new Task View as part of this review paper.

*"The discussion of many packages available to hydrologists is good. While the authors cannot discuss them all, they have a fitting selection of useful packages. For example, an analyst may be aware of the trend analysis functions in R in multiple packages, but not the spatial analysis/GIS functionality, and this article helps bring together diverse analysis methods in one discussion of the hydrology community.*

10   Thank you. Yes, indeed, one aim of the paper was to provide a broad overview of all the different tools and techniques that are available to hydrologists in R.

*"The mention of the traceability of package authors through their ORCIDs is another important, relatively recent addition to R. As an active member of the R community, I was unaware of this and found that many packages do not use this yet. However,*
15   *this is another way to judge the background of package developers and find related articles excellent information to provide to the R Hydro community."*

Yes, absolutely; we agree that adding ORCIDs to packages is a useful recent development in R!

*"The scientific quality of this article is excellent, providing a brief history of software languages used in hydrology and a brief*
20   *history of R, while making good arguments for the use of R in hydrology. Reproducible research benefits all scientific work and is an area with continued need for improvement in hydrology can help."*

Thank you for the positive comments!

*"The figures and graphics are appropriate for the article and are informative. The paper is well written and carefully edited*
25   *prior to review (much appreciated by reviewers)."*

Again, we thank the reviewer for the positive comments on the manuscript.

*" The scientific resources listed were good. It could have even been better if some of the authors were from the US and able to describe additional developments, providing a more global view."*
30   We appreciate the reviewer's opinion. The authors of this manuscript are from the UK, France, Italy, and Morocco; it originated from past contributions to the European Geosciences Union short course "Using R in Hydrology" in Vienna, which may explain why the authorship is mostly euro-centric. However, we have tried to provide a balanced, objective perspective throughout the manuscript, and we welcome suggestions if we have missed any additional developments in the manuscript.

*"A suggested improvement to the article is in the inclusion of the useR! Conference (URL provided) in the discussion of scientific resources and courses. These annual conferences are not specific to hydrology and earth sciences, but usually have some environmental science or ecology sessions as well as sessions or short courses of general interest on learning R, best practices, spatial analysis and statistical methodology incorporated in various R packages."*

Thank you, we will mention this conference in the manuscript. Many of these R conferences (RStudio, satuRdays or eRum) now stream the R talks and/or make the resources available after the event. We can certainly learn from these approaches too. Added in section 5.6.

*"Another suggested improvement is more discussion on setting up repositories. What are a few good options for hydrologists? Many subscribe to the workflow provided, but are still somewhat mystified by the repository step, that is actually rather easy once someone clears the initial hurdles of selection and setup."*

This is a very useful comment. We entirely agree and will provide a succinct overview of how to set up a repository. We also link to the Software Carpentry course which provides valuable suggestions on how to set up and structure a project repository. Added in section 3.1.

*"On page 14, lines 8 and 9, the "(Mann-)Kendall testing" punctuation is a bit awkward, exacerbated by the line break in it. I suggest leaving it as Mann-Kendall testing because readers will know what that is referring to whether they know it as Kendall or Mann-Kendall."*

Yes, thank you, we will alter this accordingly.

Done.

**Response to Reviewer 2**

We would like to thank Reviewer 2 for constructive comments on our manuscript. The comments are helpful and will certainly improve the quality of our manuscript. Below we provide the Reviewer's comments verbatim in black italic text, and our responses are immediately below each comment in blue text.

*"I enjoyed reading this paper, which is for sure useful to get an overview on what R can do for hydrologists and what hydrologists can do for R. I am an R user, and I have also developed packages, which places me at one side of the spectrum of potential readers of this paper. As such, I can benefit a lot from the information provided, especially regarding tools on subjects I am not an expert in. I wonder whether the great amount of information provided in the paper isn't overwhelming for an R beginner. It is unlikely that a R beginner will comment the paper in this discussion stage so, if I were one of the Authors, I would suggest to ask, e.g., a couple of master or starting PhD students to perform an internal review of the paper as non-expert readers (that's just an idea)."*

We thank the Reviewer for their positive comments. We find that the comment about the diverse readership of this manuscript (in terms of beginners vs. experts) is very relevant. We have thus shared the manuscript with one MSc and one PhD candidate to ask for their feedback and they both said they found the paper straightforward to follow; in fact both mentioned that the workflow figure was useful. They also raised some minor points that we will use to improve the revised version of the paper.

5    The revised manuscript will aim to make it as easy as possible for hydrologists to navigate R and the incredible amount of R resources that exist.

*"Besides the many advantages of using R (and open data and software in general), I would have expected a mention on possible drawbacks. A couple of examples follow:*

10    *- The easy accessibility and usability of open tools results, sometimes, in applications without critical scrutiny on what one does. It happened to me to receive an email from a user of my package asking me to tell him/her how to apply the methods to her/his data. A few months later I've been asked to review the paper written by the same R user who, then I realised, had no clue about the meaning of the methods applied. In other words, how can the misuse of software be reduced within the open source community? (I am aware that this happens also with proprietary software)."*

15    We feel that the misuse of software is a legitimate concern but it is indeed equally applicable to any proprietary software, as the Reviewer mentions. Recognising that it is likely beyond the scope of our paper to provide real solutions to software misuse, we will nonetheless highlight in the manuscript that one solution is to make the documentation as clear as possible. We will encourage expert users to contribute vignettes and documentation to their packages, or to write blog-posts or tutorial-style papers which explain how to correctly implement a specific method. Ensuring that use-cases are perceived as a useful

20    contribution to software and the scientific discourse is a vital step forward.
Done in section 4.1.

*"- Everybody can make an R package available, even if not carefully tested. This may result in errors that are then propagated by users. That happened to me too. I had a bug in my code but, thanks to the open source, a user spotted the error and I could*

25    *correct it. When my package was first published, I received several feedback messages that helped me a lot to clean the code. The frequency has then decreased considerably meaning that, hopefully, the main errors have been solved (or that people do not use the package any more :-)"*
*Having said this, I believe the paper is good and worth publishing in HESS, maybe after a minor revision."*
Thank you for the positive comment. We entirely agree with the Reviewer's point of view and will mention in our revised

30    manuscript both the pitfalls (risk of errors in new packages) and strengths (community review) of the open source approach. We will also place emphasis on the importance of including tests when developing an R package.
Done in section 5.1.

*"Small things:*

*Page 2, line 31: I would mention here more specifically what R is, or better what R was, i.e., software for statistics (you say that in section 2.3). This is to better inform the non-expert reader. "*

Thank you - yes indeed. We will include this information.

Done in the introduction.

*"Figure 1: I am not a big fan of this kind of figures in review papers. The text describing it could be sufficient. "*

We appreciate the reviewer's opinion but we feel this figure helps give a sense of the growth of R, providing more information than the text alone.

No change made.

*"Page 5, line 18: I would say here that hydrologic research is criticized for the lack of reproducibility, rather than scientific research in general."*

We are not sure that hydrology is worse (or better) than many other fields, however the original reference does indeed refer to computational hydrology specifically. We will thus provide a broader discussion about about lack of reproducibility, e.g. https://www.nature.com/collections/prbfkwmwvz or https://doi.org/10.1080/13697137.2018.1476968.

After careful consideration, we find that this paragraph already provides a lengthy and sufficient discussion of reproducibility, and therefore we made minor changes only.

*"Page 7, line 14: Maybe it is just the way I call things, but I would say that C and Fortran are very efficient in performing loop tasks. R, Matlab etc. are very good in vector operations."*

We tend to agree with the Reviewer: that is true in most cases (and is the reason why the `apply` functions loops are coded in C). We will remove the expression "when dealing with vector operations" from our initial sentence.

Done in section 2.4.

*"Section 2.5: Just as a note, which may be ignored by the authors if not relevant, IAHS also organises R courses in its meetings, e.g., there will be a "Using R in Hydrology" workshop at the IUGG2019 conference in Montreal. Also, among others, the PUB summer course (URL provided) uses R intensively."*

Thank you. We were not aware of these other R sessions but will definitely add this to the manuscript!

Done in section 2.5.

*"Page 10, line 16: I would not use Twitter to discover packages, but to be exposed to communications about new packages or new features. "*

Yes, we agree with this comment and will alter the text accordingly.

Done.

*"Page 12, line 24: I normally use RgoogleMaps (URL provided) to interact with Google Maps."*

Thank you, we were not aware of this but will mention it in the text!

Done.

5 *"Page 13, line 15: Actually also TUWmodel can be used in a semi-distributed fashion, allowing for differentiating into elevation zones."*

Good to know; we will include this information in the revised text. Although the functions describe a lumped model, we noticed in the examples that a semi-distributed application can be implemented. We will rephrase that sentence in the manuscript.

Done.

*"Page 14, line 19: For regional frequency analysis I would also mention the lmomRFA package (URL provided) developed by the very father of the L-moment methodology J. R. M. Hosking."*

Thank you, we will certainly include the `lmomRFA` package, as well as `lmom`, since Lmoments are important in hydrology.

Done.

**Response to Reviewer 3**

We would like to thank Michael Stoelzle for this constructive review and for his positive comments on our manuscript. We look forward to implementing these suggestions which will certainly improve the paper. Below we provide Michael's comments

20 verbatim in italic black text.

*First of all, I really support the idea of the paper and I think the presented review is fully in the scope of HESS. I like how the relationship between hydrology and R is presented and the authors gave a broad overview of historical and recent developments in the RHydro community and sketch a interesting roadmap for the future of R in hydrology. The paper is from my perspective*

25 *also a good example for a joint effort from young scientists in our hydrological community. However, since the first two reviews are full of praise I try to be a little bit more critical putting together some thoughts that hopefully improve the paper. My major "overall" suggestion is to revise the paper going a little bit away from just having a list of available packages or presenting potential possibilities in the R-Hydro universe towards a) more "best practices" for certain challenges and b) more comparisons of similar packages. However, I absolutely recommend to publish this paper in HESS. See more specific comments*

30 *below.*

*Best regards, Michael*

Thank you for these positive comments on our manuscript. We are pleased to see this work described as a "good example of joint effort from young scientists in the community", and we appreciate the thoughtfulness in these constructive suggestions. We will certainly revise the manuscript by emphasizing best practice and comparison of package functionalities, as suggested.

35 We have attempted to emphasize best practice and compare package functionalities throughout the manuscript.

*Major comments*

*The authors give a lot of technical information about R in different section in the paper (e.g. GNU S background in sect 2.3, R script workflow in Sect. 2.1, R output format in the Introduction). As all the parts are really important to learn about the*

5  *fundamental principles of R for me it would be more helpful to have the technical and historical information about R in one separate section. This might be also helpful for readers who are new to R coming from other programming languages.*

We agree it would be a good idea to move some of these elements into one separate paragraph in the introduction. However we feel that creating an entire section on the history of R might be a little misplaced, so instead we suggest to include a few general references in the introductory paragraph, such as (for example) https://cran.r-project.org/doc/manuals/r-release/R-lang.html or

10  https://cran.r-project.org/doc/FAQ/R-FAQ.html#What-is-R_003f

We have added references in this paragraph.

*I miss more discussion about different philosophies how to program in R. From my perspective there is, for example, often the question whether you are doing your visualizations with ggplot2 or in base-R or doing data wrangling with dplyr or in*

15  *a complete different "style" with data.table. I don't know if there are comparable examples for hydrological packages (doing more or less the same thing but with different techniques / packages). The paper could, however, be improved by presenting opposed approaches to solve problems in R. For example, data.table is known to be very fast in data wrangling but on the other side the dplyr package (and piping %>%) is known to support highly readable code. Here the trade-off between performance and communication could be discussed (i.e. Sect 4.3 and Sect. 2.1/2.2). My experience from teaching R in hydrology is that*

20  *students really like the concept of piping (%>% read as "and then do that") and can do the step from base-R to tidyverse in a couple of sessions. Tidyverse and Pipes could also be mentioned in Sect. 5.3 as a good possibility to teach "highly readable code" in the classroom.*

We agree that the different approaches to R programming (e.g. base-R versus the tidyverse) are definitely worth mentioning. It is often difficult to decide between different packages that offer similar options (e.g. many different packages for parallel

25  computing). The choice of approach is often quite a personal matter (i.e. depends on the individual) and many people often mix both types of approaches. Regarding which approach is easiest for beginners, we note that while the tidyverse style might seem easier at first, as soon as the analysis becomes more complex, dplyr on his own won't suffice. For example, to handle out-of-memory dplyr needs to be connected to a database, or you need to switch to bigmemory/data.table. Thus in many cases, using base-R is safer (e.g. for operational purposes, or for creating and maintaining packages) as it avoids issues with dependencies

30  and updates. Overall, we entirely agree that it is worth discussing these issues in the manuscript and providing a more explicit comparison of the different approaches - thank you for raising this point!

After considerable reflection on this point, and although we agree with Michael that the specific philosophies of R are interesting and valuable, we have come to the conclusion that they are beyond the scope of this manuscript, which aims to focus on the role of R in hydrology rather than the general use of R. Therefore, in the end we did not include this discussion, because

35  we felt it would dilute the manuscript.

*Regarding Sect 2.2 it would be helpful to draft a way forward how developers of R-Hydro packages can develop better R packages with a clear description of input data types, a comprehensive documentation on package usage. For example, tidyverse is set of packages that work well together as all packages use the same data representations and API design. Is this also a valuable approach for, let's say, developer of R packages that include hydrological models? For example, is it possible to use the data retrieval packages from Table 1 in a coherent way or is the syntax between all packages different? Is there an effort to combine data retrieval packages from different regions with a consistent R syntax/usage? Using different hydrological R packages with the same syntax might be of great value for the community. A short discussion about this issue can draft a way forward in the R Hydrology community (as mentioned in Sect 4.1 where the need of good vignettes in R is highlighted).*

Yes, we agree with the points made here. The existing hydrological data retrieval packages (e.g. rnrfa, dataRetrieval...) are structured differently because they were set up independently, and because the underlying hydrological and hydrometric datasets differ from country to country. It is therefore difficult to coordinate hydrological data amongst regions. As more nations develop similar hydrological data acquisition packages in the future, we believe it would be worth implementing a common syntax and data output form. For example, it would be ideal to use consistent APIs and consistent output objects across packages. We will make some recommendations for future developers of these hydrological packages in the manuscript. Additionally, it is worth mentioning that there is currently no effort to combine data retrieval packages from different regions but this is certainly worth doing. Perhaps the World Meteorological Organization's Hydrological Observation System (WHOS) might consider implementing (http://www.wmo.int/pages/prog/hwrp/chy/whos/index.php) something like this - for example, (i) a meta-package with functions that convert hydrometric data from other packages to a standard format, or (ii) a meta-package for all hydrological models to be run within the same framework.

We have included this discussion in sections 5.1-5.3., including a discussion of data standards (e.g. Open Geospatial Consortium WaterML)

*The introduction of section 3 and also Fig. 3 are really nice and well-conceived. I like the idea to split the data analysis into different parts and subsections in the paper (Sect. 3.x).*

Thank you!

*In general, the paper is full of valuable package, links, and corresponding citations. However, from my perspective often only the package names are given and more detailed information how the package is working or why it is especially useful/important for the hydrological community is missing. Of course, it is not possible to describe all the functionality of all the packages in detail but the authors could pick 2-3 cases where they really compare the advantages and disadvantages of using different packages for the same purpose / challenge. To be honest (and I also think that the author team is aware of that), you can relatively easy check on the internet if there is a package for a certain task and what are the technical details or data requirements of a single package. So, the great list of available packages cannot be the main message of this compelling paper! Comparing different packages among each other could really be of great value for the hydrological community and this paper is a good*

*place to do that. A good example is Sect 3.4 describing spatial data analysis and visualization in R. In this section a lot of important R packages are mentioned but the reader gets less information which package is better or has the same functionality as other packages etc. In Sect 3.5 the information about snow functionality in the hydrological models is a good example how this can be done.*

5  Yes, it is true that in some cases we were quite brief - there is so much to say that this paper could be written as a book! However, we agree that it is worth comparing the advantages/disadvantages of similar packages when we revise the manuscript. We will provide some information about advantages of specific packages within the different sections, similarly to the snow functionality example.

We have included discussion of the merits of specific packages at various points throughout the manuscript (for example, a
10  comparison between `mgcv` and `gamlss` in the statistical section, or the discussion of different statistical modelling packages in the models section. We are unable to do this for all packages because this is already a very long manuscript and would turn into a book if we did!

*For me a major deficiency of the paper is that the topic "colours" is completely missing. Hydrology is a multi-facetted discipline*
15  *and hydrologists doing a lot of visualizations for posters, presentations and papers. Hydrology is also often based on multi-dimensional analyses with different variables in space and time. To visualize hydrological processes and hydrological change colour is often the first choice to compare data sets and to investigate relationships. As we all know, a sophisticated colour choice is often a huge challenge. I encourage the authors to implement a "colour section" where a short discussion about, for example, the basics of appropriate colour gradients for graphs and maps is given (one-colour gradient, two-colours gradient,*
20  *discrete vs. continuous variables). In large-scale hydrology "rainbow" colour gradients are still state-of-the-art (to generate fancy and colorful world maps) but a lot of papers and blogpost have taught us that the "rainbow" isn't perceptual uniform and not colourblind-safe.*

Indeed, there is a lot to say about data visualization - we agree this would be a useful addition. It is widely accepted that the rainbow color scheme is a poor choice for data visualization (e.g. https://betterfigures.org/, https://www.climate-lab-book.ac.uk/
25  2014/end-of-the-rainbow/, or a recent R-bloggers post on this topic - https://www.r-bloggers.com/at-the-end-of-the-rainbow/). We will mention that R is strong in this department and will include a short section about colours, color-blind friendly palettes, and appropriate choices in hydrology. We will also mention some useful papers on this subject, e.g. **?**.

Done in section 3.7.

30  *Minor comments*

*P2L20 Give 1-2 examples of hydrological models coded in R.*

Yes, we will do this.

Done. We discussed TOPMODEL, FUSE, TUWmodel, and various other models in section 3.5 rather than in the introduction.

*P2L33-35 It might be helpful for the reader to compare the fundamental principles of R against other programming languages (e.g C, Java) to better describe the architure of R (e.g. in terms of object-orientantion, complilation of code*

Yes, we will include a short description of R's principles (as an object-oriented programming and interpreted language) in the technical section mentioned above.

Done in section 2.1. instead.

*P3L4 Are there any reasons for the large number of updates in R-Hydro packages (related to Fig.1)?*

We actually had a lengthy discussion about this among ourselves. One possibility is that the increase in updates in 2018 is related to documentation requirements on CRAN, as a couple of features that were okay before had to be modified (possibly due to package dependencies). Alternatively it is possible that the rise in package updates was linked to the development and uptake of R-HUB (a web service that allows developers to test and debug R-packages on different operating systems to reproduce what CRAN does). We were not entirely sure of the answer and so we refrained from mentioning this in the manuscript.

No action taken.

*P4L11 The platform Stackoverflow (from Sect. 2.5) can also be mentioned here as there often also information on relevant R packages are given.*

Yes, we will do this.

Done in section 2.5. We were careful to avoid redundancies in the manuscript.

*P5L31 What is meant with the "standardized format"? Here it might be also valuable to add packages like roxygen2 that help to develop, create and document R packages.*

The sentence referred to is *"Relying on well-established publishing platforms such as CRAN and GitHub has promoted a standardized format for developing and disseminating R code."* Here, we were referring to the best practice in writing R code, and will clarify this in the text. We will include the roxygen2 package.

Done.

*P6L27 It will be easy to find some justification for this statement (e.g. number of R topics on stackoverflow.com compared to other programming languages).*

The statement referred to is *"R is still considered the most powerful language and environment for statistical computing"*. We will justify the statement in our revised manuscript.

Done.

*P6L3-14 The paper would be improved by adding various possibilities to share and publish code during and after paper publication (e.g. github repository, gists or other tools and platforms). Is there a possibility to share R code with a DOI?*

We agree, this is a good idea. We will mention different optionsand platforms for sharing R code with a DOI such as GitHub, Zenodo, Figshare, and RPubs/Plotly (also for dashboards and interactive plots). Another important data repository to mention is https://www.pangaea.de/, because it is tailored to Earth and Environmental Sciences. Some of these platforms are discussed in the journal *Geoscientific Model Development* https://www.geoscientific-model-development.net/about/code_and_data_policy.

5   html

Done (new section 5.1).

*Sect 2.3 For me this section contributes less to the strength of the paper – what is the key message here? The specific statistical packages that are relevant for R-Hydro are given in Sect 3.6 (as mentioned). The link to CRAN is perhaps too unspecific here.*

10   *Yes, R provides a vast number of statistical tools for hydrologists but then you have to explain in more detail why and what advantages R here has to offer (compared to other languages?).*

Yes, there is indeed some overlap between sections. We will remove this opening section (*2.3. Providing statistical tools for hydrology*) and strengthen section 3.6 (*Packages for hydrological statistics*) instead. We will move the description of CRAN to the new suggested section that provides a background on R.

15   In the end we decided that there is merit in keeping the brief discussion of the importance of statistics in R in the opening sections of the manuscript, and keeping it separate from the discussion of statistical packages later in the manuscript. We did remove the description of CRAN.

*Sect 2.5 What are the rotating topics and why have they been chosen? The picture might be nice to see but the space could*

20   *instead better be used to highlight the EHGU SC topics from the last years with a short information why the topics were of great interest for the community. It might be also important to highlight the possibility to raise issues for certain R packages on github.com to foster an online-documented discussion with the package developers (i.e. interaction in the community). At least a cross reference to Sect. 5.4 could be made here.*

The section Michael is referring to (Section 2.5.) is the section on *scientific resources and courses*. We will keep the picture

25   and will include a list of the topics too, explaining why they are of interest. We also agree that it is worth highlighting in this section the ability to raise issues with the developers on GitHub (provided packages are developed/published on GitHub). We will encourage publication of packages on GitHub and will mention that the bugs report link can be added in the Description file so that package authors can specify how they prefer to receive bug reports.

We added the cross-reference to 5.6. Raising issues is mentioned in section 5.2. The short course topics are already listed in

30   5.6.

*Table 1 I am not sure on which basis you have choose the 11 packages? Are they the most important ones? Perhaps a thematic order would improve the table (sort by data type, spatial extend, multiple proposes or not,...)*

Yes, we agree that ordering these packages by data type (and perhaps providing a bit more information about the data) would

35   be worthwhile.

Done.

*P11L5-9 Consider to mention the rio package here as it is really powerful. The tidyverse package is cited as a package that reads multiple file formats – I think this is not the main purpose of the tidyverse package.*

Thank you for the suggestion about the `rio` package. We will indeed rephrase our description of `tidyverse`!

Done.

*P11L10 Which netcdf package is the best? Or let's say, are there more information which package to use for a specific application.*

We will discuss which package is most appropriate for different uses.

After reflection, they perform similar roles and we feel it is not our place to make that judgment, so have not added more information here.

*P12L15 For all ggplot2 users it is really important to mention that the ggplot2 3.0 version offers support to visualize sf objects directly with a specific geom (geom_sf). So, if one is familiar with ggplot2 and want to visualize spatial data (raster, shapefiles etc.) this is a really easy way to do that? As the ggplot2 community is growing and growing this might be a really important information for some readers of the paper.*

This is a great idea; thank you for the suggestion.

Done.

*Table 2 Should be included in Sect 3.5*

Yes; we will rectify this.

Actually it already was in section 3.5, but the LaTeX formatting made it look like it wasn't.

*Sect. 3.6 Package "skimr" might be valuable for this section.*

Good idea, thank you!

Done.

*P15L10 I agree, one strength of ggplot2 and faceting is the ability to generate multiple views. This is especially important to avoid overplotting or – as the authors mentioned – to split a data set in different plots using meta information of this data set (i.e. season). If you have a nice example for faceting here would be a good place to show this principle for the readers.*

We do have some good examples and will include one, thank you!

Done.

*P19L17 Is it true that publishing a R package on CRAN is free of quality checks? However, a short comment on the differences during package publication (CRAN vs.GitHub) might be helpful here.*

In our experience, CRAN only checks for code and documentation consistency, not code logic nor quality/coverage (CRAN does not run linter checks or test coverage). The point about the differences during package publication is important and discussed briefly in section 5.2.

Done.

*P19L21-26 I agree but the paragraph could be more specific on how we should develop R packages. Is there a good article or blogpost describing the progress of publishing a R package? What are the requirements to generate good tutorials or readme files? Might it be useful if journals like HESS recommend to publish paper code on github?*

We will include (1) a description of the process, or provide a clear, straightforward reference on publishing packages; and (2) a description of what a useful tutorial/readme should include for hydrology. Some references for package building include, for example: (i) a popular minimal description of how to write a package https://hilaryparker.com/2014/04/29/writing-an-r-package-from-scratch (ii) helpful tutorials such as https://kbroman.org/pkg_primer/ or a more tidyverse oriented reference with a video: https://www.rstudio.com/resources/videos/you-can-make-a-package-in-20-minutes/. We will also suggest that journals should encourage authors to publish their code. For example, the journal *Geoscientific Model Development* already does this: https://www.geoscientific-model-development.net/about/code_and_data_policy.html.

Thank you for the helpful suggestions.

We have included the reference by Hadley Wickham, which is the clearest possible description of how to create an R package.

We have discussed vignettes in section 4.1. There is also already a discussion encouraging authors to publish their code.

**References**

[revised manuscript text omitted]

[Note: We have created a new section here, inserted some of the text from 5.2, restructured the paragraph and added information about sharing platforms following Reviewer 3's suggestion]

[revised manuscript text omitted]